# Ancient DNA reveals diverse community organizations in the 5th millennium BCE Carpathian Basin

Anna Szécsényi-Nagy [1] ✉, Cristian Virag [2], Kristóf Jakab[1], Nadin Rohland[3,4], Harald Ringbauer [5], Alexandra Anders [6], Pál Raczky [6], Tamás Hajdu [7], Krisztián Kiss [1,8], Tamás Szeniczey[7], Sándor Évinger [9], Tamás Keszi [10], Zsuzsanna M. Virág[11], Olivia Cheronet [12,13], Swapan Mallick [3,4,14], Ali Akbari[3,4,14], Ron Pinhasi [12,13], David Reich [3,4,14,15] ✉ & Zsuzsanna Siklósi [6,16] ✉

Little is known about the genetic connection system and community organization of Late Neolithic and Early Copper Age populations of the Carpathian Basin. Here, we present a comprehensive genetic investigation of these populations, leveraging whole genome data from 125 individuals. Using population genetics, kinship analyses and the study of networks of identity-by-descent haplotype segment sharing, we elucidate the social and genetic dynamics of these communities between 4800–3900 calibrated years BCE. Despite changes in settlement patterns, burial practices, and material culture, we document a high degree of genetic continuity. While one set of individuals from a large community cemetery is genetically diverse, another site is more homogenous and closed, with numerous consanguineous relationships and evidence of patrilineality and patrilocality. In this work, we document important differences in kinship systems in contemporaneous Early Copper Age communities using similar material culture and living only about 100 km apart.

The geographical position of the Carpathian Basin makes it a crossroads between South-East and Central Europe. The fertile river valleys of the Danube and its tributaries offered optimal conditions for settlement and facilitated the development of efficient transport and communication networks for early farming communities. Since the advent of the field of aDNA research, the prehistoric populations of the Carpathian Basin have been the subject of intensive study, primarily due to the critical role they played in the Neolithisation of Central Europe[1,2]. The population genetic history of the Neolithic (6000–4500 calibrated BCE) of the Carpathian Basin is well-established[3–6]. However,

[1]Institute of Archaeogenomics, HUN-REN Research Centre for the Humanities, 1097 Budapest, Hungary. [2]Satu Mare County Museum, 440031 Satu Mare, Romania. [3]Department of Genetics, Harvard Medical School, 02115 Boston, MA, USA. [4]Broad Institute of MIT and Harvard, 02142 Cambridge, MA, USA. [5]Department of Archaeogenetics, Max Planck Institute for Evolutionary Anthropology, 04103 Leipzig, Germany. [6]Institute of Archaeological Sciences, ELTE Eötvös Loránd University, 1088 Budapest, Hungary. [7]Department of Biological Anthropology, Institute of Biology, Faculty of Science, ELTE Eötvös Loránd University, 1117 Budapest, Hungary. [8]Institute of Practical Methodology and Diagnostics, Faculty of Health Sciences, University of Miskolc, 3515 Miskolc, Hungary. [9]Department of Anthropology, Hungarian Natural History Museum, Hungarian National Museum Public Collection Centre, 1083 Budapest, Hungary. [10]Hungarian Natural History Museum, Hungarian National Museum Public Collection Centre, 1083 Budapest, Hungary. [11]Budapest History Museum, Aquincum Museum, 1031 Budapest, Hungary. [12]Department of Evolutionary Anthropology, University of Vienna, 1090 Vienna, Austria. [13]Human Evolution and Archaeological Sciences Forschungsverbund, University of Vienna, 1090 Vienna, Austria. [14]Howard Hughes Medical Institute, 02138 Boston, MA, USA. [15]Department of Human Evolutionary Biology, Harvard University, 02138 Cambridge, MA, USA. [16]MTA-ELTE Lendület "Momentum" Innovation Research Group, 1088 Budapest, Hungary. ✉e-mail: szecsenyi-nagy.anna@abtk.hun-ren.hu; reich@genetics.med.harvard.edu; siklosi.zsuzsanna@btk.elte.hu

the subsequent Copper Age (4500–2800 cal BCE) is underrepresented in the available genetic data[4,7–10].

The Copper Age is characterized by the spread of significant technological innovations, including metallurgy[11,12] and the wheel and wagon[13,14], which profoundly impacted the later history of Europe. This period is thought to coincide with the emergence of salient social ranking and craft specialization[15,16].

During the Late Neolithic (4900–4500 cal BCE), tells and large horizontal settlements were established in the Carpathian Basin[17], with some covering an area of up to 60-80 hectares (ha). These sites were distinguished by a concentration of population[18–20]. The number of inhabitants of the Late Neolithic settlements has been estimated to be several thousand based on the number of excavated buildings and burials[19,21]. The use of these extensive settlements terminated by the end of this period, between 4500 and 4450 cal BCE[22]. Profound transformations were observed on the Great Hungarian Plain (GHP), affecting all segments of life. At the beginning of the Copper Age, in contrast to the large horizontal Late Neolithic settlements, a dense network of small, farm-like settlements emerged across the GHP[23,24]. During the Late Neolithic, the deceased were buried within the settlement boundaries or through rites that left no archaeologically visible traces[25]. However, in the Early Copper Age, from 4400–4350 cal BCE, formal cemeteries – the first in the Carpathian Basin – that were spatially separated from the settlements – appeared[22]. Meanwhile, the practice of settlement burials persisted[26,27], with even Late Neolithic tells being repurposed for burial grounds[28,29]. These changes coincided with shifts in the material culture, particularly in the pottery style. Subsequently, on the GHP, the Tiszapolgár[30] and then the Bodrogkeresztúr style emerged[31,32] (Supplementary Notes 1-2, Supplementary Fig. 1). In parallel, in Transdanubia (Western Hungary, west of the Danube River), the pottery style of the Lengyel complex underwent a transformation[33,34]. The analysis of grave goods reveals disparities in wealth, while the use of prestige and status goods provides evidence of social differentiation[25,35]. It has been proposed that the Early Copper Age formal cemeteries functioned as a kind of common burial place, a central place for the inhabitants of several nearby smaller settlements[22,36].

The interpretations of these transformations are diverse, encompassing a range of hypotheses. These include migration[37], economic changes[38], climate change[39], and internal social reorganizations[40]. However, some elements of cultural continuity between the Late Neolithic and Early Copper Age of the GHP are more supportive of an internal, peaceful transformation[27]. This interpretation is reinforced by anthropological and strontium isotope analyses, which indicate no significant increase in individual mobility or evidence of a new immigrant population between the Late Neolithic and the Early Copper Age[41,42].

We report genome-wide data from 125 Late Neolithic and Early Copper Age burials to address two interrelated questions (Fig. 1, Supplementary Data 1). First, we test whether a population shift occurred between the Late Neolithic and Early Copper Age, and if so, whether such a shift correlates with the observed cultural and economic transformations. Secondly, we aim to understand the organization of the Late Neolithic and Early Copper Age communities, specifically whether we can discern evidence of internal reorganization related to the changes between the two periods. To investigate these questions, we conduct a comparative analysis between two significant Late Neolithic archaeological sites, Aszód-Papi földek and Polgár-Csőszhalom, and two large, almost completely excavated Early Copper Age cemeteries, Tiszapolgár-Basatanya (referred to as Basatanya in this paper) and Urziceni-Vamă, which exhibit a high degree of cultural similarities to one another. The Polgár microregion, situated on the Upper Tisza River, provides an ideal setting for investigating local continuity. Extensive excavations in this 91 km² area have uncovered several archaeological sites spanning from the Middle Neolithic

(Polgár-Ferenci-hát (5470–5070 cal BCE)) through the Late Neolithic (Polgár-Csőszhalom) to the Early Copper Age (Polgár-Nagy-Kasziba and Basatanya)[43]. To provide a broader regional perspective, we incorporate human remains from additional contemporaneous Copper Age sites, including those from Budapest-Albertfalva-Hunyadi János út, Iváncsa-Lapos, and Rákóczifalva-Bagi föld site 8 (Supplementary Note 1, Supplementary Data 2).

There are few studies of genetic relatedness of prehistoric periods in Hungary[6,10], although previous analyses of other areas of Europe document a variety of kinship and residence practices[44–48]. To obtain insight into internal transformations affecting the organization of communities that underlie archaeologically visible cultural changes, we combined population genetic methods with biological relatedness analyses[49,50] and identity-by-descent (IBD) segment sharing analysis[51].

In this study, we show population continuity at the transition between the Late Neolithic and the Early Copper Age on the GHP, despite cultural changes. Accompanying this phenomenon, Early Copper Age communities became more isolated, and first-cousin marriages became a common practice, at least on the Northern GHP. However, this cannot be generalized on the level of archaeological culture, as at least one culturally similar, contemporaneous community on the Eastern margin of the GHP did not follow the same trend. The differing internal organization and genetic connection patterns of the Basatanya and Urziceni-Vamă cemeteries suggest that the later might have been a common burial place for a larger community with more distant connections, while the Basatanya cemetery served a relatively closed, tightly interrelated and probably smaller community.

## Results
We generated genome-wide data by shotgun sequencing and capturing more than 1.2 million single nucleotide polymorphisms (SNPs) in 125 individuals who lived during the Late Neolithic and Early Copper Age of the Carpathian Basin, and report enhanced capture ($n = 6$) and shotgun genomes ($n = 8$) of individuals for which aDNA data were published previously[4,5,7] (Fig. 1). We combine the data reported here with previously published data from the Carpathian Basin[4–7,52] and other genomic data from relevant ancient Europeans (see Methods).

As different regions were reached by new technological innovations at different times, there is a variation in the chronological nomenclature across the regions surrounding the Carpathian Basin. We use the terminology specific to the given region and provide corresponding radiocarbon dates of the major reference groups in Figs. 1–2. Archaeological cultures used for classification should not be interpreted as coherent social groups that necessarily existed in the past[53–55]. Behind the diversity of material culture lies a range of social identities, and the extent to which material cultures correspond to coherent social groups must be tested and evaluated on a case-by-case basis rather than assumed from the outset. To avoid the pitfall of conflating archaeological cultures with specific populations, we use individuals and communities as the basic units of analysis, and test whether there is any correlation between the use of various pottery styles and biological relatedness.

### Genomic composition of the studied communities
We carried out a principal component analysis (PCA) on genome-wide data using *smartpca*[56], projecting ancient individuals onto modern Western Eurasian genetic variation to visualize key trends in both the study period and area (Fig. 2A). Our Late Neolithic data, combined with previously published Neolithic genomes from the Carpathian Basin[4,6,57], is located on the PC1-2 space between Neolithic Anatolia, Western Europe, and northeastern (Bohemian) Late Neolithic groups, hinting at moderate hunter-gatherer (HG) ancestry, that we demonstrate in later analyses. Within the Carpathian Basin, GHP communities show higher HG ancestry compared to their Transdanubian counterparts[4]. On the GHP, the Late Neolithic period showcases a more homogeneous

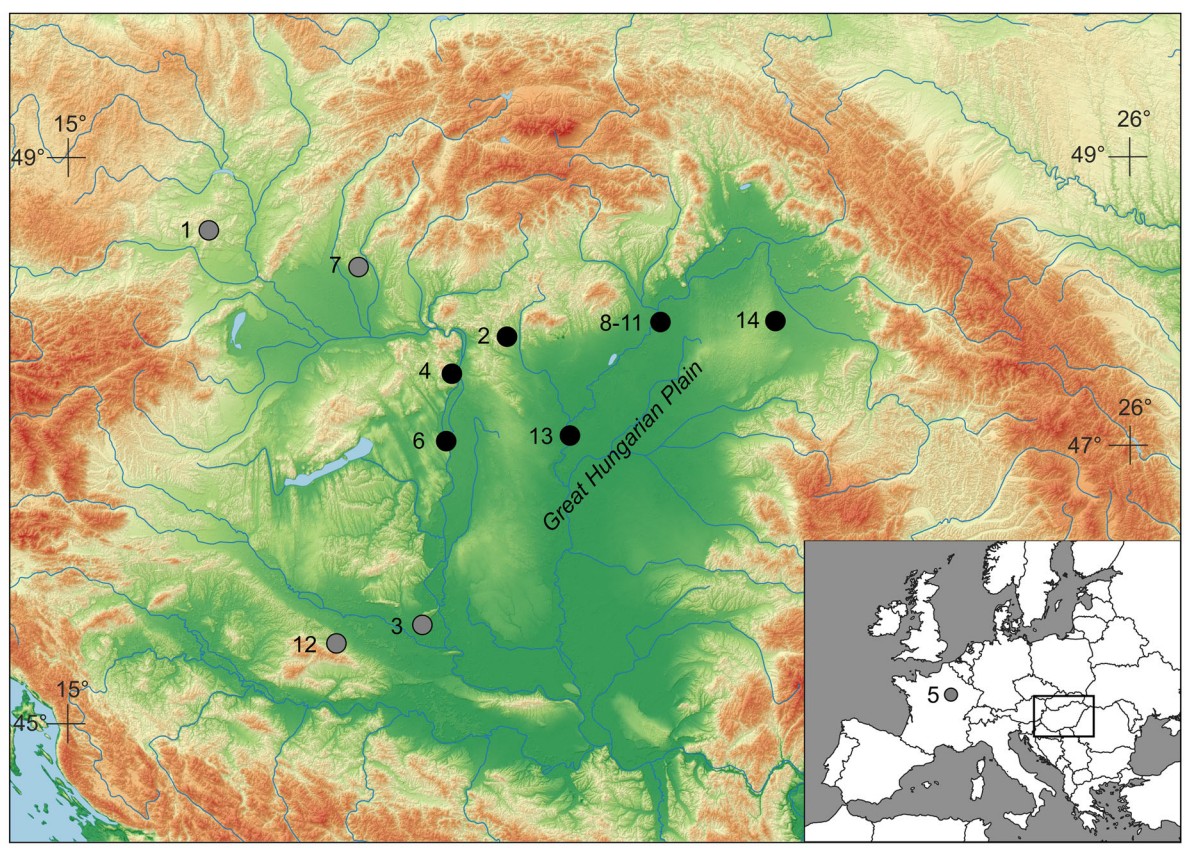

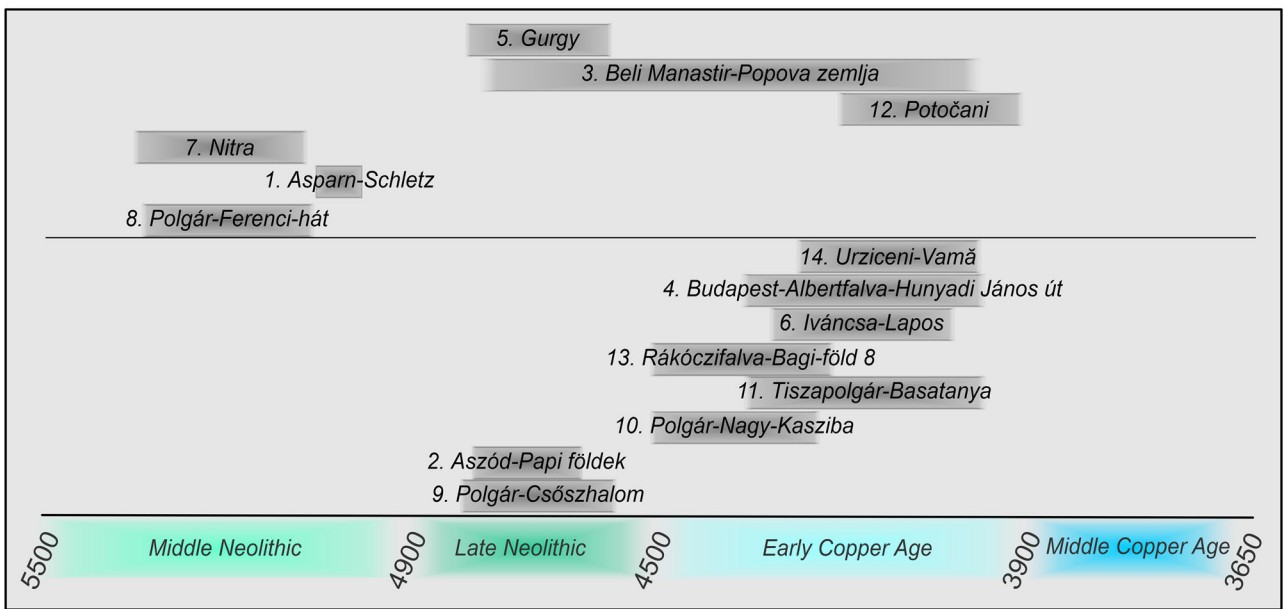

**Fig. 1 | Map of the Carpathian Basin with the sites mentioned in the text and their chronologies.** Date range is given in gray bands at the bottom in cal BCE. Black markers indicate sites for which we generated ancient DNA data, gray markers indicate separately published comparative data. Numbers 8–11 cover the sites from the Polgár microregion. Basemap was made with Natural Earth (free vector and raster map data @ naturalearthdata.com) and SRTM provided by NASA (https://doi.org/10.5067/MEASURES/SRTM/SRTMGL1.003).

ancestry pattern than the Middle Neolithic (see Supplementary Data 3 for Levene's and Bartlett's tests for homogeneity). The Early Copper Age populations of the Polgár microregion display greater homogeneity and tighter clustering (p < 0.05, Bartlett's test, Supplementary Data 3), where only two PCA outliers are identified. In contrast, contemporaneous Early Copper Age individuals from Urziceni-Vamă (Romania) and others from Romania and Bulgaria[58] exhibit signs of increased ancestry diversity, reflecting new Eastern European genetic contacts. Using Mahalanobis distance and testing based on a chi-square distribution, we detect in Urziceni-Vamă five female and three male outliers along different PC axes (p < 0.05, Supplementary Data 3), with some individuals shifted toward the Eastern European steppe, and others toward Northwestern or Anatolian populations.

We used qpAdm software[59–61] to model samples using Anatolian Neolithic farmer (ANF), and Western and Eastern European HG (WHG and EHG) as sources, to estimate admixture proportions in the Late

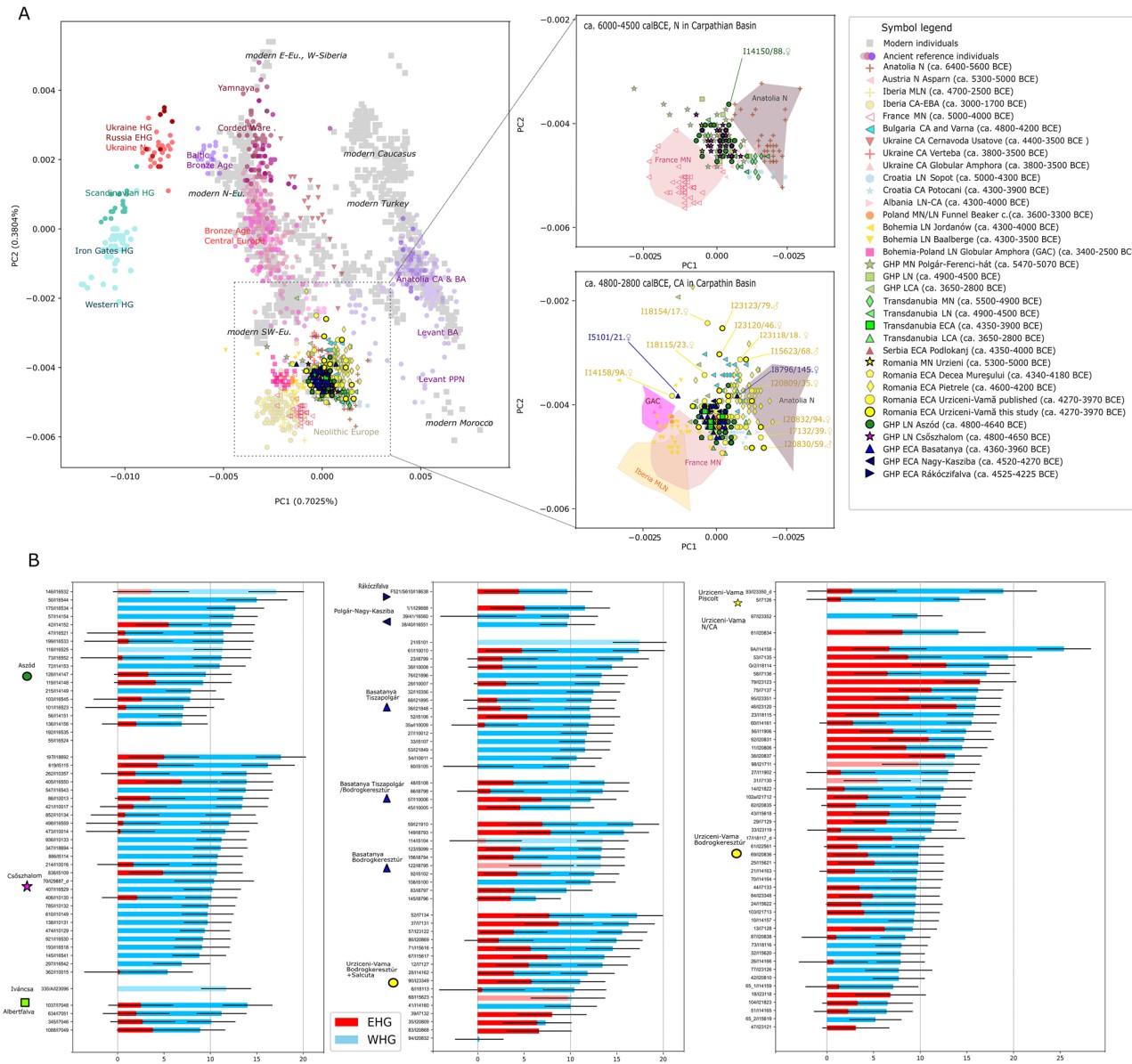

**Fig. 2 | A: Scatterplot showing PC1-2 of a principal component analysis, B: Individual ancestry inferences based on a 3-way qpAdm model. A** Among the individuals in this study, outliers (all >50 k SNPs detected, tested by Mahalanobis distance calculation combined with χ² distribution, at α = 0.05; see Methods and Supplementary Data 3) are marked on the zoomed-in parts of panel A. The two zoom-in panels focus on the Neolithic and Copper Age samples from the Carpathian Basin with comparative data, whose chronology is indicated in the figure legend. Abbreviations in the legend: N Neolithic; MN Middle Neolithic; LN Late Neolithic; MLN Middle–Late Neolithic; CA Copper Age; ECA Early Copper Age; LCA Late Copper Age; EBA Early Bronze Age; GHP Great Hungarian Plain. Source data for

the scatterplot are provided in the Source Data file. **B** Error bars represent qpAdm estimates as centers with one standard error, calculated using the block-jackknife approach implemented in qpAdm (see Supplementary Data 5A). Individual genotypes were the units of study, used as targets in qpAdm tests. Abbreviations are as follows: WHG Western Hunter-Gatherer; EHG Eastern Hunter-Gatherer. The Anatolian Neolithic ancestry component is omitted from the chart. Grave numbers and laboratory IDs are indicated on the individual barplots. Faint bars denote qpAdm tests with p < 0.05, indicating a lack of fit. For further acceptable models, p-values and component estimates, see Supplementary Data 5.

Neolithic–Early Copper Age genomes presented here (see Methods, Supplementary Data 5A–F); the inclusion of multiple hunter-gatherer groups reflects an attempt to cover the diversity of HG ancestry contributing to people of this region[4,7]. In our analyses, as expected, ANF is the dominant component in all Neolithic to Copper Age populations of the GHP (88% on average, Fig. 2B, Supplementary Figs. 2–3). The distribution of EHG and WHG components shows limited EHG in the Late Neolithic GHP communities on one end, and more in Early Copper Age Urziceni-Vamă on the other (a two-sample t-test of their pairwise comparison resulted in p < 0.002 for the EHG component variance Fig. 2B, Supplementary Data 5A, B).

Both the Late Neolithic Aszód-Papi földek and Late Neolithic Polgár-Csőszhalom populations can be modeled as direct descendants from the earlier local Middle Neolithic Polgár-Ferenci-hát population without additional admixture, using qpAdm (p = 0.26 and 0.40 respectively), consistent with their similar position in PCA. We tested ancestry sources of the subsequent Early Copper Age GHP individuals, and modelled 24 out of 30 individuals at Basatanya using the Polgár-Csőszhalom community as a single source (p > 0.05, Supplementary Data 5E). At Urziceni-Vamă, 28 out of 50 Early Copper Age individuals can be modeled using a local Middle Neolithic source alone; this lower proportion of fitting models suggests additional admixture into this

community from exogenous sources (Supplementary Data 5E). Similarly, only half of the Early Copper Age individuals fit as derived without admixture from a population genetically identical or closely related to Late Neolithic Polgár-Csőszhalom ($p > 0.05$). Consistent with the ancestry variability observed in $f_4$-statistics (Supplementary Fig. 4), Early Copper Age Urziceni-Vamă individuals have evidence of several streams of post-Neolithic influx and show diverse ancestry proportions, which, however, do not correlate with pottery styles (such as Bodrogkeresztúr- and Salcuţa-styles, see Supplementary Fig. 1, Supplementary Data 4–5). Sporadic evidence of CHG (Caucasus hunter-gatherer) ancestry is evident in graves 17, 18 and 31, which can be early signs of steppe influence into the area, since CHG + EHG appears jointly, like in later Yamnaya-related groups[7]. However, in nine cases, CHG can be fit without an EHG component ($p > 0.05$, with component Z-score>2 Supplementary Data 5D). In a group-based qpAdm analysis, the genetic composition of the Urziceni-Vamă is most similar to Basatanya, whereas the comparative Copper Age Romanian Pietrele and Bulgarian Varna site groups show significant differences in their WHG composition[58] (Supplementary Data 5F). This reflects the observation that the Varna group has genetic components from all WHG, EHG and CHG in similar proportion (4–6%), alongside the predominant ANF ancestry. Although the Varna population is not a suitable group source for Urziceni-Vamă overall, five Urziceni individuals who do not fit the Late Neolithic GHP source can be modeled using either Varna or Pietrele as a single source ($p > 0.05$, Supplementary Data 5E).

The Tiszapolgár-style graves at the Basatanya site show a non-significantly lower EHG component compared to those that have Bodrogkeresztúr-style pottery (ca. 2.6% on average compared to 4.6%, two-sample $t$-test $p = 0.0765$, see Supplementary Data 5B). There are also relatively few genetic outliers at Basatanya, with only three individuals failing the ANF-WHG-EHG 3-way qpAdm model test (Supplementary Data 5A).

## Parental relatedness and effective population size

To obtain insights into the population composition and effective population size ($N_e$) of the studied groups, we first inferred and analyzed runs of homozygosity[62] (ROH). We compared the genome fraction in ROH with previously published Neolithic and Copper Age datasets[6,8,44,63,64]. An excess of larger ROH segments (>20 centimorgan or cM) at the Polgár-Ferenci-hát site provides evidence of sporadic close-kin unions, which are absent in other contemporaneous Neolithic groups of Central Europe[6,63] (Fisher's exact test with Polgár-Ferenci-hát vs the combined LBK sites yield a $p$-value = 0.035, Fig. 3A, Supplementary Data 6A). In the subsequent Late Neolithic Aszód-Papi földek and Polgár-Csőszhalom communities, contrasting on a microregional level to the Middle Neolithic phenomenon, there is evidence of only one close-kin union on the order of 2nd-cousin level (5th degree) parental relatedness. In the following Early Copper Age, two individuals from Urziceni-Vamă and six out of 30 studied individuals at Basatanya have significantly increased levels of long ROH (sum >50 cM in segment >20 cM) indicative of consanguineous relationships between 3rd-degree relatives, such as first cousins (see details on Supplementary Fig. 5). This is notably high in ancient datasets (Basatanya vs. sum of ten reference groups: $p$-value = 0.0001, see Supplementary Data 6A).

Both at Basatanya and Urziceni-Vamă, pottery styles do not show a correlation with signs of inbreeding, leading us to conclude that the practice of consanguineous unions was not associated with distinct pottery stylistic traditions of the sampled communities. It is, however, noteworthy that the outlier individuals observed on the PCA bear no or only limited (4-8 cM) ROH signals.

These shorter observed homozygous segments (4-8 cM) attest to background relatedness, typical of populations with low recent effective population size[62] ($N_e$). We calculated the $N_e$ based on the length distribution of 4-20 cM ROH (likely arising from co-ancestry mostly within the last ~50 generations) across comparative Neolithic and Copper Age populations. We also calculated diploid $N_e$ based on IBD segment sharing of high quality genomes (see Methods, Fig. 3B, Supplementary Data 6). During the Middle Neolithic-Copper Age periods, we track fluctuations in hapROH $N_e$ over time in the Polgár microregion, and show ~41–63% larger mean $N_e$ in Urziceni compared to Basatanya and Polgár-Csőszhalom (Supplementary Data 6B). However, we observe a decrease in effective population size of the Polgár area Early Copper Age communities in the IBD-based $N_e$ calculations (by approximately a quarter at Basatanya compared to the Late Neolithic). This decreasing trend of IBD-$N_e$ signals appears to be restricted to the Polgár microregion, as compared to Basatanya it is 6–7× larger at Urziceni-Vamă and 2-3× higher in the mass grave from Potočani (Croatia), both belonging to contemporaneous Early Copper Age communities (Supplementary Data 6C).

As a comparison, we added the well-studied Neolithic Gurgy (France) dataset to the $N_e$ analyses, which is understood as a patrilocal population with controlled female exogamy, where most pairs of parents are related to each other via co-ancestors within the preceding 5–30 generations[44]. The population stability recorded there, with the relatively even distribution of the ROH segments and extensive pedigrees[44], stands in contrast to what we observe on the GHP. When comparing the ROH patterns of Gurgy with the two well-recorded communities of this study, we find that Basatanya's ROH patterns are more varied, including both individuals without ROH and others with longer homozygous segments. Consequently, community organization seems to have been unstable at the Late Neolithic–Early Copper Age period of the Carpathian Basin, when the ancestors of the Basatanya population experienced a relatively recent effective population decline (apparent in an abundance of 12–20 cM ROHs, and numerous IBD sharing), most plausibly reflecting a more closed community organization at Basatanya compared to the more open system at Urziceni-Vamă.

## Genetic network of Neolithic-Copper Age individuals

To assess the genetic structure and the genetic network of the Late Neolithic–Early Copper Age groups in relation to other contemporaneous populations, we inferred and analyzed long shared haplotypes, commonly referred to as IBD segments, based on phased and imputed autosomal haplotypes using the ancIBD software[51] (see Methods).

Using all individuals with adequate quality data (see Methods) and applying a filter for at least 12 cM shared segments between two individuals, the ForceAtlas2 algorithm[65] produces a dense graph of the Carpathian Basin Late Neolithic-Copper Age samples showing scattered IBD sharing with Neolithic populations of France, Poland and Bohemia and a more intensive network with Middle Neolithic and Copper Age individuals in Croatia (Fig. 4A).

The degree centrality ($k$), representing the number of links held by each node, calculated from the network shown in Fig. 4A, averages 0.077 for Basatanya–the highest value among the sites in this study (Supplementary Data 7)–placing it in a central position within the network. Despite including individuals from only 19% of the excavated graves, Basatanya exhibits nearly three times more within-cemetery IBD connections than Urziceni and twice as many as Csőszhalom ($k$w, Supplementary Data 7A). Nevertheless, it is also linked with the contemporaneous nearby Polgár-Nagy-Kasziba site and the Late Neolithic population of Polgár-Csőszhalom. All analyzed individuals from these three sites – except graves 45 and 83 from Basatanya and 197 at Csőszhalom – form a single cluster based on Leiden community detection applied on the same network[66], including 17 individuals from the Urziceni-Vamă cemetery embedded within this cluster (Supplementary Fig. 6, Supplementary Data 8). Analyzing the sites further as modules of the network, we demonstrate that other

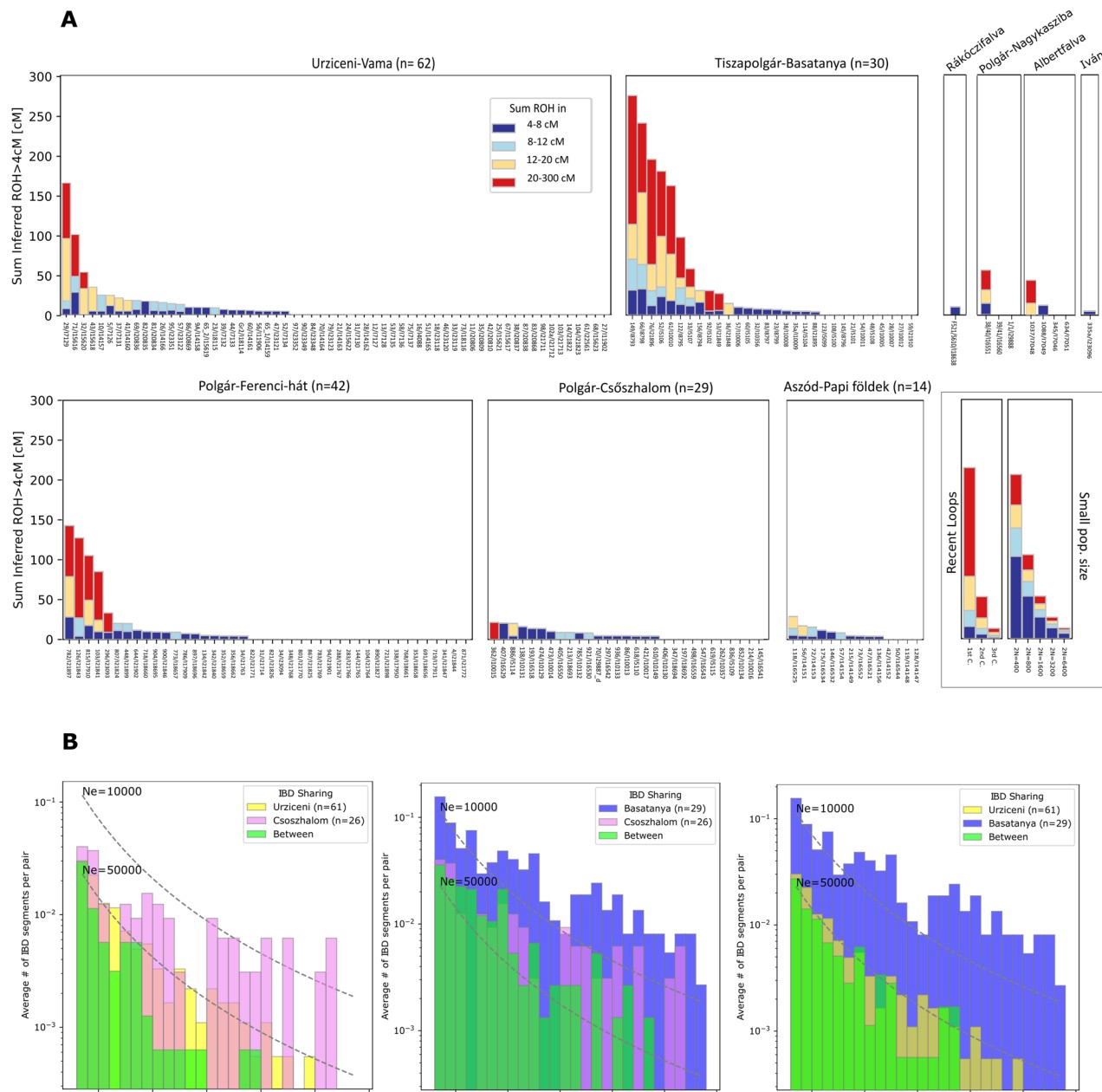

**Fig. 3 | Inferred Runs of Homozygosity (A) and pairwise identity-by-descent (IBD) patterns (B) of the studied sites' populations. A** Plot of ROH for each individual, summing them in the form of stacked, colored bars, depicting different ROH lengths. Individuals from the same sites are grouped into boxes, indicating the number of individuals suitable for ROH analysis (>400k 1240k SNPs covered) in the title of each box. In a separate panel, the legend is depicted with expected ROH for offspring of close kin in an outbred population ("Recent Loops") and panmictic populations of different sizes, using the calibrations from Ringbauer et al.[62]. We observe signals of consanguinity between close kin in two cases at Urziceni, six cases at Basatanya and four cases at Middle Neolithic Polgár-Ferenci-hát[6]. Signs of

small effective populations are detectable in shorter ROHs. Further in-depth analyses are seen in Supplementary Fig. 5 and Supplementary Data 6, exact data are seen in Source Data file. **B** Inter and intra site IBD sharing patterns and IBD-based effective population size estimates, using TTNe.analytic script (see Methods). The plots demonstrate the larger effective population sizes of the Urziceni and Csőszhalom communities compared to Basatanya. The third IBD plot shows that inter-site IBD connections between Basatanya–Urziceni occur at the same rate as within-site connections at Urziceni, indicating that Urziceni population is so much larger that individuals from it are no more closely related to each other than they are to Basatanya individuals.

Late Neolithic and Copper Age genomes from present-day Hungary and the majority of the Urziceni-Vamă community exhibit less dense within-module clustering than Basatanya and Csősz-halom (Supplementary Data 7A). Although, 48% of the graves excavated at Urziceni-Vamă are part of the IBD analyses, some Urziceni-Vamă individuals (graves 9 A, 23, 18, 35, 39, 68) show no connections to anyone on the graph or appear as outliers within their community (Fig. 4A), a pattern that is consistent in four

cases with the PCA outlier detection as well (Supplementary Data 3).

### Late Neolithic community organization and the Late Neolithic–Copper Age transition

Twenty-nine individuals (16 females and 13 males) have data of sufficient quality for IBD analysis from the Late Neolithic Polgár-Csőszhalom site. They show high within-site (within-module,

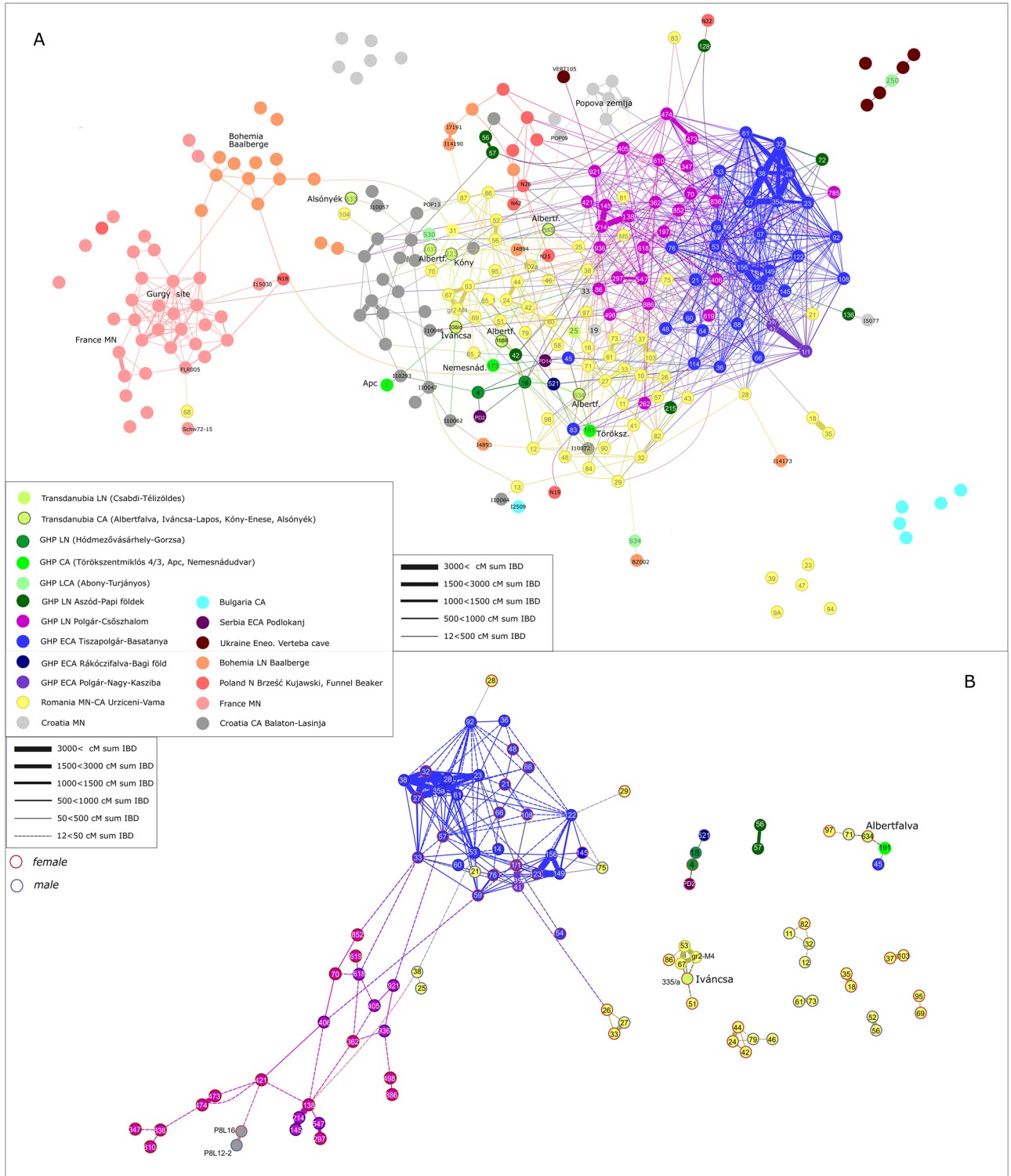

**Fig. 4 | Network of IBD segments shared between the individuals from the Carpathian Basin and other roughly contemporaneous Neolithic and Copper Age individuals of Europe.** Source data for the network are provided as a Source Data file and Supplementary Data 8. ForceAtlas2 layout is used in Gephi software v0.10.1 for both graphs (see Methods), which considers the number of connections (centrality) of each node (genome) and where edges (IBD connections) between nodes act like springs. The total length of IBD sharing is used to weight the edges as indicated in the legend. Thicker edges represent closer relatives, including familial relationships between individuals, while thinner edges depict more distant connections. Numbers in the nodes indicate grave numbers at the given site in this study, while reference samples are labeled by lab codes as provided in AADR[122] v54.1. Abbreviations in the legend: N Neolithic; MN Middle Neolithic; LN Late

Neolithic; MLN Middle–Late Neolithic; CA Copper Age; ECA Early Copper Age; LCA Late Copper Age; GHP Great Hungarian Plain. **A:** Network of sharing 1 × 12 cM IBD segments between individuals from the Carpathian Basin and other roughly contemporaneous Neolithic and Copper Age individuals of Europe. **B:** At least 2 × 12 cM IBD segments connectivity shown between the Late Neolithic and Copper Age individuals in the Carpathian Basin. Genetic sex is indicated by the outline colors of circles. These networks illustrate how interconnected the Carpathian Basin individuals were with each other and their adjacent regions (**A**), and how this network, under stricter filtering, fragments even communities from the same sites, while still keeping Basatanya, Polgár-Nagy-Kasziba and Polgár-Csőszhalom connected, zooming into the GHP area (**B**).

$k = 0.05$) connectivity, but even stronger between-site (between-module) connections ($kB/k = 0.732$, representing the ratio of between-module connections to total connections, Supplementary Data 7), documented also by the maximum IBD segment length (strength) ratios (Supplementary Figs. 7–8). We measured cliques within the network as complete subgraphs—subsets of at least three nodes where each pair of nodes is adjacent, forming a complete subgraph. Clique analysis reveals that Polgár-Csőszhalom individuals are more likely to participate in between-module cliques (103 cliques) compared to within-module cliques (24 cliques), with females predominantly driving these inter-site connections ($p = 0.0015$ in Chi$^2$ test, Supplementary Data 7-9, Supplementary Fig. 13). Three small kindreds identified at Polgár-Csőszhalom are detailed in the Supplementary Note 3 (Supplementary Fig. 9). Individuals from Polgár-Csőszhalom maintain links in the IBD network with the Early Copper Age Basatanya cluster, even under strict filtering ($2 \times 12$ cM; Fig. 4B). This continuity is observed through three males (graves 406, 618, 921) and one female (grave 852), suggesting local continuity in the Late Neolithic–Early Copper Age transitional period. The three youngest graves (graves 421, 498 and 619) at Polgár-Csőszhalom do not represent direct connections with Basatanya, suggesting that the interaction between the populations occurred earlier than the end of the Polgár-Csőszhalom site. The maternal and paternal haplogroup frequency distribution at the two sites is consistent with maternal continuity ($p = 0.421$), while the paternal composition differs significantly ($p = 0.022$) (Supplementary Figs. 10-11, Supplementary Data 10).

## Early Copper Age community and cemetery organization on the Great Hungarian Plain

We sampled 14 females and 16 males from 154 Early Copper Age graves at Basatanya (Supplementary Fig. 12C, see Methods). Among them, 29 are suitable for IBD analysis. The Basatanya population stands out with the highest degree centrality, clustering coefficient, and within-module strength among the three major GHP sites, indicating a highly interconnected community (Supplementary Data 7). Many of these individuals are part of extended pedigrees, with close biological relatedness reflected in long stretches of IBD (details in the next sections). Nevertheless, a significant portion of the population also maintains between-module connections (Supplementary Figs. 7–8).

A distinctive feature of the Basatanya community is its resilience in the IBD network, even when stricter filter is applied (requiring at least $2 \times 12$ cM or longer for network inclusion; Fig. 4B). The Leiden algorithm also connects individuals with different pottery styles, with only graves 83 and 45 being excluded from the cluster (Supplementary Fig. 6, Supplementary Data 8). Mapping the weighted within-cemetery $2 \times 12$ cM IBD connections, we observe a high level of interconnectedness among most individuals (average degree is 7.24, graph density is 0.259), with the exception of those buried in graves 83 and 45 (Fig. 5). Although the genomic and uniparental genetic make-up of these two IBD outlier males is not unusual for the time or region, their IBD connection patterns point outside the site, toward Urziceni-Vamă and Early Copper Age populations on the southern GHP and Transdanubia. The male in grave 83 ha no detected ancestor or descendant in the community, whereas male 45 has a single 12 cM IBD connection to an individual in grave 36. Their limited ROH signals also suggest that they were likely outsiders, differing from the typical closed-community pattern seen in the Basatanya community. However, their grave goods are similar to the rest of the burials. Other individuals with high ROH are scattered throughout the cemetery, without any clear association to family ties or pottery styles (Fig. 5). Despite these two male IBD outliers, analyzing the 72 cliques observed at Basatanya, we find more female connections between sites, whereas males predominantly participate in cliques within the Basatanya community ($p = 0.0036$, Chi$^2$ test, Table 1, Supplementary Fig. 13C).

Through a combination of READ, KIN, ROH, and IBD analyses (see Methods), we identified three biological kindreds at Basatanya, two of which represent larger pedigrees (Fig. 5), linking graves in relative proximity in different sections of the cemetery. Family A, located in the western section, is a complex, extended kin group buried with Tiszapolgár-style pottery. Through READ and KIN analyses, we identified the nuclear family consisting of a mother (grave 27), a father (grave 28), a son (grave 35a), and a daughter (grave 32). IBD and KIN results further indicate that the individual in grave 23 is in second-degree ancestral relation to grave 28 and third-degree paternal relative of the male in grave 35a. The Y-chromosome lineage I2a1b1a(2b)-L1316/Z161 (ISOGG v15.73) can be traced across at least three generations in this pedigree, and is detected in precursor Polgár-Csőszhalom community too. This lineage is present in 10 out of the 16 males analyzed at Basatanya site (Supplementary Fig. 14). Additionally, another paternal lineage, C1a2b-Z44491, appears across two distant generations within this pedigree. In contrast, four distinct mitochondrial lineages are found in Family A (Fig. 5, Supplementary Fig. 10, Supplementary Data 1).

Family B is associated with Bodrogkeresztúr-style burials and includes a mother (grave 123) and her two sons (graves 149 and 156), along with three more distant relatives (graves 59, 76, and 122), with grave 122 representing a descendant through the maternal line. Interestingly, grave 76 contains Tiszapolgár-style inventory, linking the two pottery styles through biological relatedness. We find a marked difference in ROH between the two siblings in graves 149 and 156, with 149 likely being the offspring of close relatives (probably 3rd-degree related, e.g. first cousins, as seen in Supplementary Fig. 5). Taking into account the significant variability in ROH expected from simulations[62], the observed IBD/ROH patterns, and our kinship analyses, 156 could also be an offspring of the same parents (see Supplementary Note 3).

Compared with strontium ($^{87}Sr/^{86}Sr$) isotope data[42], we find that all males in families A and B have, on a microregional scale, local strontium values, while the individuals in graves 114 and 145, who have no close biological relatives, exhibit non-local strontium isotope values in their first molars; thus the genetic and isotopic analyses converge in showing these individuals to be outliers. The uniparental, genetic relatedness, IBD clique analysis and the strontium evidence all speak for a predominantly patrilocal and patrilineal community organization (Table 1).

The Polgár-Nagy-Kasziba site is clearly embedded within the Basatanya and Polgár-Csőszhalom IBD cluster, with individuals in graves 1/1 and 39/41, displaying significant inter-site connections. We sampled a nuclear family here, consisting of a mother (38/40), a father (39/41), and their 3–4 year-old daughter (1/1). The daughter is a 6th–8th-degree relative of a male in grave 23 at Basatanya, as revealed through IBD analysis (Fig. 4B and Supplementary Fig. 16). Both father and daughter show multiple IBD connections to individuals in Basatanya, making it challenging to determine the directionality of the connections between these chronologically and geographically close sites (Supplementary Note 1, 3).

From the Early Copper Age Urziceni-Vamă cemetery, we sampled 43 females and 34 males (Supplementary Fig. 12D, Methods), with 62 individuals among them suitable for IBD analyses (47.3%). Despite the substantial representation of this cemetery – from an almost completely excavated site – it forms the most fragmented IBD network, with the fewest internal connections among the three major GHP sites (see Methods, Fig. 4A, Supplementary Data 11). The average number of connections within Urziceni-Vamă itself is 2.96, with a graph density of 0.125, which is significantly lower than the density of the connections of the contemporaneous Basatanya (-1/4 of that, Supplementary Data 7). In the $2 \times 12$ cM IBD network, many connections disappear (Fig. 4A vs. B), leaving predominantly 1st-2nd-degree relationships dominating the second network, and only a few Urziceni individuals remain connected to the Polgár-Csőszhalom and Basatanya cluster.

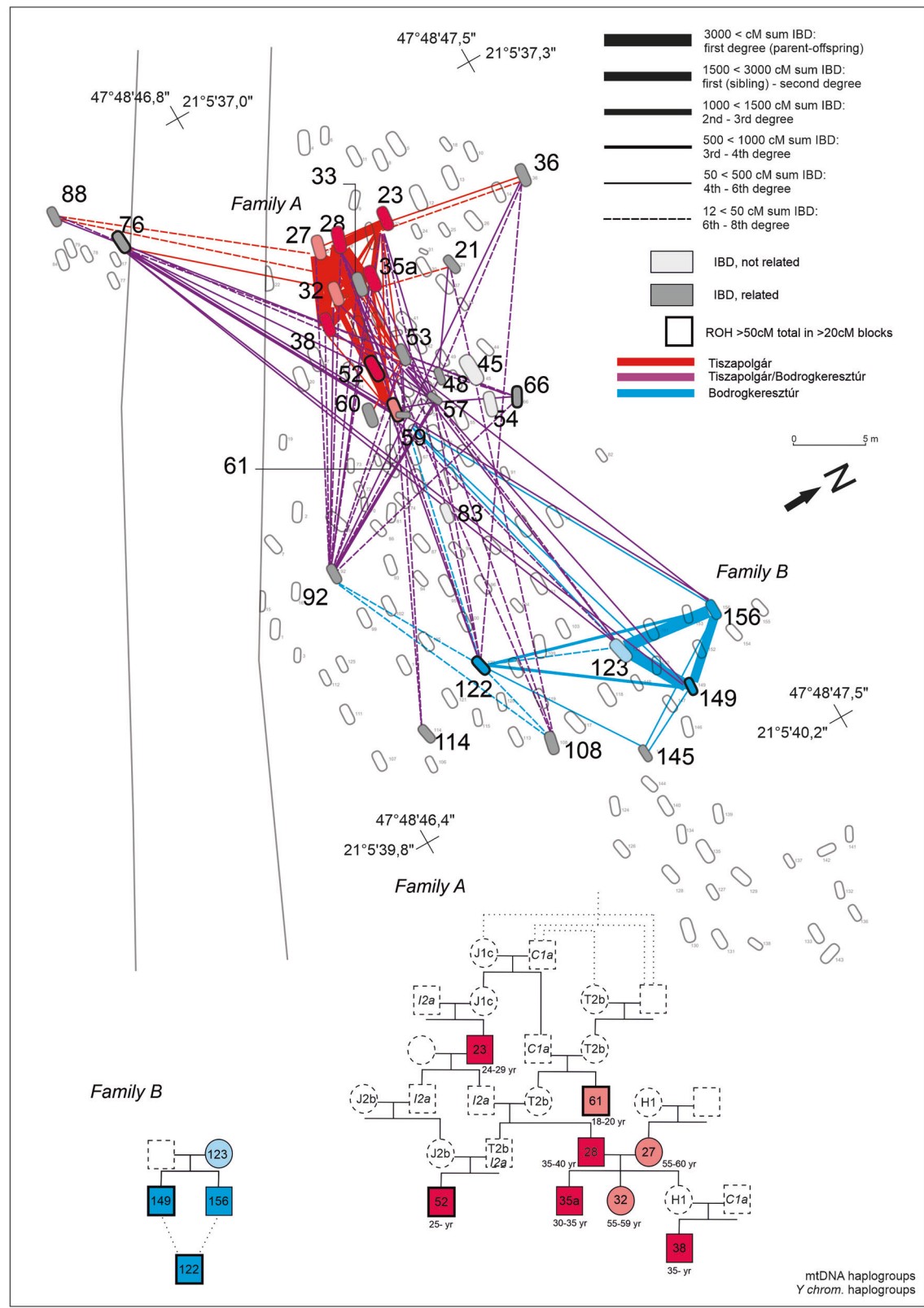

Eight close biological kindreds are detected at Urziceni-Vamă in the IBD analyses, with three additional kindreds identified through READ and KIN analyses, which allow study of individuals with lower quality data (see Methods and Supplementary Data 1). These pedigrees consist of two to three individuals, with most connections involving adult siblings, and no evidence of biological fathers. Despite the presence of close relatives, kindreds do not form substructures on the IBD graph, mapped onto the cemetery, except for three brothers from graves 4, 53, and 67 (Fig. 6). This is also reflected in the IBD metrics, where some individuals have high between-module strength, but their degree centrality remains low (Supplementary Figs. 7-8).

Eighteen cliques are counted within Urziceni-Vamă, with four including children, which is ~1/5 of the between-module cliques observed at this site (79 total). Importantly, both male and female

**Fig. 5 | Map of the Tiszapolgár-Basatanya cemetery with biological relationships between the buried individuals.** All pairs with at least 2 × 12 cM identity-by-descent (IBD) connections are marked on the cemetery map. Edges are weighted according to the total length of IBD sharing, as indicated in the legend. Colors of the edges represent different pottery styles: Tiszapolgár (red), Bodrogkeresztúr (blue) and lines connecting the two (violet). Family A and B are represented as pedigrees. The pedigree was reconstructed based on KIN, READ, IBD, hapROH analyses, as well as mitochondrial DNA and Y haplotype sharing. Black outline highlights graves and individuals in the pedigrees bearing at least 50 cM sum IBD of ROH over 20 cM in length. Source data are provided as a Source Data file and Supplementary Data 1, 11,

12. Color shades of the individuals indicate local $^{87}Sr/^{86}Sr$ signal (vivid color) or lack of Sr isotope data (faint symbols). The presented pedigree of Family A is the simplest version explainable by the results. Besides this, several further versions could be conceivable, considering the missing family members. Family A and B have also further distant (3rd+ degree, such as graves 59 and 145 in Family B) members with uncertain positions in the pedigree that are shown by IBD connections on the cemetery map. In addition to the presented Families A and B, third-degree relatives are identified using KIN (in graves 57 and 92) in Family C. IBD results suggest that the relatives have an avuncular 2nd or ancestral third generation relationship (Supplementary Fig. 16).

**Table 1 | Comparison of the two Early Copper Age communities through the most important metrics and characteristics**

| | Tiszapolgár-Basatanya Early Copper Age | Urziceni-Vamă Early Copper Age | Significance in difference between the sites |
|---|---|---|---|
| **Anthropological sex estimation (in excavated graves)** | 43.0% male, 35.76% female, 19.9% children, 1.3% symbolic | 27.9% male, 43.3% female, 21.2% children, 8% not determined | |
| **Bayesian-modeled radiocarbon dating of the cemetery (68.3%)** | 4355-4270 to 4150-3990 cal BCE | 4285-4130 to 4035-3940 cal BCE | |
| **Pottery styles** | Tiszapolgár, Bodrogkeresztúr | Bodrogkeresztúr, Salcuța | |
| **% ind. analyzed in PCA (n analysed / excavated)** | 19.5% (30/154) | 55.8% (72/129) | |
| **% ind. analyzed in IBD (n analysed / excavated)** | 18.8% (29/154) | 47.3% (61/129) | |
| **% PCA outliers** | 6.7% (2/30) | 11.1% (8/72) | $p = 0.49$ |
| **% ind. with EHG Z score > 2 in qpAdm** | 3.3% (1/30) | 24.6% (16/65) | $p = \mathbf{0.026}$ |
| **IBD kw/k** | 0.433 | 0.313 | $p = 0.86$ |
| **IBD cliques W/B** | 0.98 (423/432) | 0.49 (50/102) | $p = \mathbf{0.0001}$ |
| **IBD cliques Chi² W/B module male/female diff.** | $p = \mathbf{0.0036}$ | $p = 0.2289$ | $p = \mathbf{1.79 \times 10^{-5}}$ (males W/B) |
| **IBD $N_e$** | 6000-7500 | 38,900-50,900 | 6.65× diff. |
| **% Y-chr diff. lineages/ total** | 18.8% (3/16) | 36.7% (11/30) | $p = 0.357$ |
| **% diff. mtDNA subhaplogroups/ total** | 60% (18/30) | 58.7% (44/75) | $p = 1$ |
| **% Inbreeding (min 50 cM, > 20 cM blocks)** | 20% (6/30) | 3.3% (2/61) | $p = \mathbf{0.014}$ |
| **Cemetery organization** | 3 kindreds with 2-6-8 members, related inds. buried in proximity of each other | 11 kindreds with 2-3 members only, buried also in larger distances, no fathers in pedigrees | |
| **Community structure** | patrilineal, patrilocal | non-patrilineal, non-patrilocal | |

*P*-values were calculated from Z-test, two-tailed test, Fisher's exact, and Chi² tests. Individual data of this table and calculations are presented in Supplementary Data 1, 3, 5, 6, 7, and 10. *p*-values below the 0.05 threshold are highlighted in bold.

individuals exhibit high between-site connections, with no significant difference in connectivity between sexes ($p = 0.23$, Table 1, Supplementary Fig. 13C). Interestingly, two females in a mother-child relationship (graves 18 and 46) joined the community from external groups, carrying also elevated EHG components (9% and 14%, respectively).

After subdividing the Early Copper Age Urziceni-Vamă dataset based on pottery styles (Bodrogkeresztúr and the mixed Bodrogkeresztúr-Salcuța (Supplementary Note 2, Supplementary Fig. 1), we observe significant differences in network connectivity. Individuals buried with Bodrogkeresztúr-style grave goods ($n = 48$) form within-style cliques (seven cliques with 22 nodes), while individuals ($n = 14$) with Bodrogkeresztúr-Salcuța-style grave goods do not form within-style cliques. The ($k_B/k$ *style*) ratio is significantly larger for the individuals in Bodrogkeresztúr-Salcuța-style graves compared to the values of the individuals in Bodrogkeresztúr-style graves in line with the style specific pattern of the cliques ($p < 0.001$ in Fisher's exact test, Supplementary Data 7A). Notably, pottery styles do not strictly align with family structures, as demonstrated by mixed-style cliques (five cases) and the appearance of Bodrogkeresztúr-Salcuța-style pottery in only one grave per families C, E, F, J and K. Although the available radiocarbon data do not allow for a chronological separation of

the two styles, the archaeological chronology, pedigrees and age at death of certain individuals support the version A of the family trees F and K. This phenomenon might indicate that the Bodrogkeresztúr-Salcuța-style pottery appeared in the cemetery rapidly, within a time span of one or two generations.

The uniparental composition reflects the site's broader and looser network. A total of 44 different mitochondrial DNA subhaplogroups were identified (Supplementary Fig. 10, Supplementary Data 1), including rare haplogroups pointing toward the southeast, such as N1b (based on its rarity in the area and era and its general appearance in the Near East[67,68], and others (e.g. J2b1, J1c) with parallels to the Varna and Pietrele sites[58]. The paternal genetic diversity is also larger at Urziceni-Vamă than at Basatanya (Supplementary Fig. 11). Y-lineages include C2a, which is associated with Mesolithic people and H2 and G2a elements of Neolithic origin[1], as well as potentially new external influences such as R1b from steppe-outliers (graves 12 and 79).

## Discussion

Our analyses show that the genetic ancestry of Late Neolithic GHP communities derives from the preceding Middle Neolithic

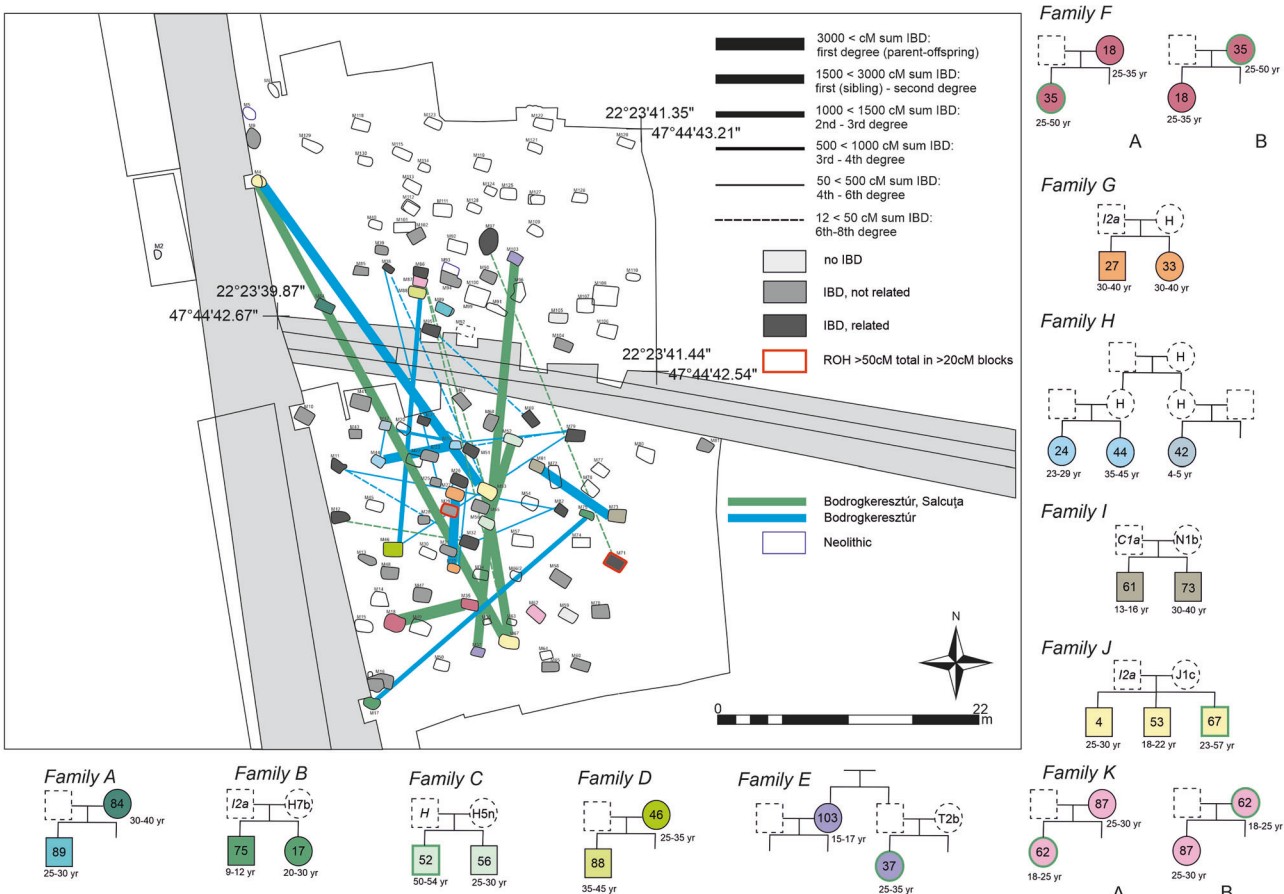

**Fig. 6 | Biological relatedness in the Urziceni-Vamă population.** At least 2 × 12 cM identity-by-descent (IBD) connections are marked on the cemetery map. Edges are weighted according to summed IBD, as indicated in the legend. Source data are provided as Source Data file, Supplementary Data 1, and 11. Colors of the edges represent IBD connections among individuals buried with Bodrogkeresztúr (blue), Bodrogkeresztúr-Salcuța (green) pottery styles. Pedigrees are shaded both on the map and the pedigree consistently. The pedigrees are reconstructed based on KIN,

READ, IBD analyses and mitochondrial DNA and Y-chromosomal lineage sharing (Supplementary Data 12). Highlighted graves and individuals on the map bear at least 50 cM sum IBD of runs of homozygosity (ROH) over 20 cM in length. The presented pedigree of Family E and H are consistent with the results, but several alternative versions are possible, considering the missing family members. Green outlines of the symbols in the pedigrees signal Bodrogkeresztúr-Salcuța-style pottery in the grave.

populations, with a noticeable decline in genetic diversity over time. These populations have minimal Eastern Hunter-Gatherer (EHG) ancestry, aligning with the genetic trends described by Lipson et al.[4]. During the Early Copper Age, however, we observe differences in the genetic composition among the GHP sites. In the Polgár microregion of Hungary, $f_4$-statistics, qpAdm, and IBD analyses (Figs. 2 and 4) consistently indicate that the cultural changes between the Late Neolithic and Early Copper Age occur without the introduction of larger immigrant groups. The Early Copper Age population is predominantly descended from local Late Neolithic communities, sharing mostly maternal lineages and exhibiting even greater genetic homogeneity, which likely indicates increased isolation. Only two adult female individuals (graves 21 and 145) from the Basatanya cemetery are identified as outliers in their genetic ancestry composition. Still, their burial rites and grave goods do not differ from the other graves. These unrelated individuals likely joined the local community during adulthood, and became connected to it through their descendants (see IBD analyses). In contrast, contemporaneous Early Copper Age sites, such as Urziceni-Vamă, exhibit greater genetic diversity (PCA, $f_4$, qpAdm) and stronger connections to Eastern European regions. Similar patterns are observed at Pietrele and Varna in southeastern Europe[58]. Interestingly, most of the outliers at Urziceni-Vamă are female ($n = 5$ out of 8), and their varied ancestry does not correlate with pottery styles, burial practices, or grave goods.

Evidence suggests that individuals from the Volga Basin region may have sporadically appeared in the Early Copper Age of the GHP, as shown by the burial at Csongrád-Kettőshalom in Hungary, dated to 4330−4070 cal BCE[69,70]. However, these cases appear to be isolated in the currently available dataset. For instance, the Decea Mureșului cemetery from Romania, which culturally reflects an early infiltration of Eastern steppe groups as well, yielded genetic results for one analyzed male (grave 10) that are similar to those of early farming populations of the GHP[71].

The IBD networks reflect the diverse internal organizations of the studied communities. We observe that closely related individuals were buried near one another at both Late Neolithic Aszód-Papi földek and Polgár-Csőszhalom (Supplementary Figs. 9 and 15). While similar grave goods are noted (Supplementary Data 2), the small number of detected biological connections makes it challenging to assess whether these similarities are due to kinship or mortuary customs tied to local, age, or gender-specific practices. Females at Polgár-Csőszhalom play a significant role in connecting different sites, suggesting extensive external relationships, as indicated by the high number of inter-site IBD connections. Polgár-Csőszhalom is most closely connected to Basatanya, though separated by a 300 year chronological gap within the dataset, sharing long IBD segments with Basatanya and having more connections with other sites than within itself (Fig. 3B). We detect 6th-8th-degree or more distant relations on maternal lineages

of three males and one female at Polgár-Csőszhalom to all families at Basatanya (Fig. 4). As these individuals at Polgár-Csőszhalom are not among the chronologically youngest graves, we interpret this evidence as some ancestors of the Basatanya population diverged earlier from the Csőszhalom community than the Polgár-Csőszhalom site was abandoned.

In the Late Neolithic period, the effective population size ($N_e$) is uniformly large across the studied sites (Supplementary Data 6). In contrast, the Early Copper Age presents a more complex scenario. While $N_e$ remains large or increases at Urziceni-Vamă and Croatian Potočani, the Polgár microregion experiences a significant $N_e$ decline, as shown by the IBD $N_e$ analysis. A relatively large $N_e$ suggests the presence of extensive marriage networks at Urziceni-Vamă, while a 6.6-fold smaller $N_e$ could imply a more isolated, closed community at Basatanya. Whether this contraction is also accompanied by population decline on the GHP, remains to be verified through further sampled Early Copper Age groups.

The closure of Early Copper Age communities in the Polgár microregion is also supported by the numerous third-to-fourth-degree kinship connections within the Basatanya cemetery (Supplementary Data 11-12, Supplementary Fig. 16). This pattern likely extends beyond a single site and reflects a broader microregional trend, exemplified by strong IBD ties between Basatanya and the nearby Polgár-Nagy-Kasziba site (located ~10 km away, Figs. 1, 4, 5), indicating a highly interrelated and connected community system. At Basatanya, despite the use of various pottery styles and the identification of at least three pedigrees, the IBD network reveals a strongly interrelated cemetery community. In terms of social organization, female individuals are not generally exogamous; however, IBD analyses show that they play a key role in linking sites across the region, while males are primarily responsible for intra-site connections at Basatanya (Supplementary Data 7, Supplementary Fig. 13). This, along with the uneven distribution of the uniparental lineages, the eight-member family structure, and the stable isotope evidence, points to a patrilineal and patrilocal society at Basatanya (Table 1).

A long-term comparative analysis of human populations over the past 15,000 years indicates that inbreeding has been relatively rare. Two distinct forms of inbreeding can be identified: one resulting from genetic drift in small populations, and the other due to consanguineous unions[72]. Inbreeding due to small population size was more prevalent in Paleolithic and Mesolithic hunter-gatherer groups and in isolated island communities[73], though studies suggest that deliberate avoidance of consanguinity was common[74]. As Neolithic population growth reduced the likelihood of inbreeding, its occurrence became even rarer. When it did occur, it was typically within farming communities under conditions of strict social stratification, elite isolation, and the inheritance of private property[46,75–77].

The prevalence of first-cousin marriages varies considerably across cultures, spanning a spectrum from complete prohibition to socially preferred practices in modern societies. A longitudinal analysis indicates that the preference for first-cousin marriages is a relatively recent phenomenon, influenced by various social factors, including the intergenerational transfer of assets[72,78–80].

In the case of the Early Copper Age Basatanya cemetery, the elevated rates of ROH indicate first-cousin unions. Similar trends are observed in Malta and the Aegean islands, where the islands' isolation likely contributed to inbreeding patterns[46,73]. In contrast to these examples, the Basatanya community did not experience such geographic isolation, as archaeological evidence points to a densely populated network of small settlements. However, the contemporaneity and precise relationships between these settlements and cemeteries remain unclear. This suggests that the isolation of the Basatanya community was more likely socio-cultural in nature rather than geographical.

A conscious strategy to preserve and transfer wealth within the family lineage could be one potential explanation for the prevalence of consanguineous unions in Family A at Basatanya. However, we do not observe a clear correlation between biological relatedness and the wealth of burials, in the analysis of individuals sharing long IBD segments and those who do not. When considering the grave goods of Family A across four generations and those of Family B, it is possible that access to prestige goods from distant areas gradually diminished, or the family lost interest in them as the community became more isolated. This is evidenced by the absence of such items in the graves of Family B members (Graves 123, 149, and 156) and in the younger generations of Family A (Graves 32, 35a, and 38). Another possible interpretation is that only one child inherited the family's property or rank, as evidenced by some richly furnished burials at Basatanya (Fig. 5, Supplementary Data 2, Supplementary Note 1). Furthermore, the limited pool of potential mating partners may also have contributed to the occurrence of consanguineous relationships. However, since only 19.5% of the cemetery has been analyzed, these interpretations remain speculative until further analysis is conducted on a larger sample. In summary, the Basatanya cemetery appears to reflect a highly cohesive, patrilineal community, where first-cousin marriages were common over generations. The burial tradition of placing spouses (graves 27-28) and close relatives near one another echoes similar patterns observed in the Middle Neolithic[6] and Late Neolithic periods of the region.

The Early Copper Age Urziceni-Vamă cemetery presents a strikingly different social structure compared to the Basatanya cemetery. Despite sharing partially identical and partially similar burial traditions, it was used by a more open and probably larger community. One key observation is the lack of direct lineal relationships spanning more than two generations, while sibling relationships are common. Fathers are notably absent from parent-child burial relationships, and even closely related individuals are often buried far apart from each other. One potential explanation for this phenomenon is that fathers were buried elsewhere, in their natal community's cemetery in a matrilocal system. In five cases, Bodrogkeresztúr-Salcuța-style pottery appears alongside Bodrogkeresztúr-style pottery within small pedigrees, but no pedigrees are found where all members used Bodrogkeresztúr-Salcuța-style pottery exclusively. This phenomenon may indicate that the Bodrogkeresztúr-Salcuța-style was introduced into the cemetery quickly, within two generations.

Consanguinity at Urziceni-Vamă was rare, contrasting with the frequent first-cousin unions seen in Basatanya, indicating that cousin marriages were not a common practice at Urziceni-Vamă. We found no clear correlation between biological relatedness and grave goods. The small kindreds and absence of strong biological ties between individuals buried near each other suggest a community not organized around strict patrilineal rules. Although it remains a possible explanation that the Urziceni-Vamă cemetery was used for a considerably shorter time than the Basatanya cemetery or that it reflects a neolocal community, the currently available AMS (accelerator mass spectrometry) dates suggest otherwise. The cemetery likely remained in use for 140−300 years at a 68.3% probability, a timeframe that aligns closely with the Basatanya cemetery's estimated duration of 160−325 years at a 68.3% probability, based on Bayesian-modeled radiocarbon dates (Supplementary Figs. 17–20).

The differences in $N_e$ observed between the Late Neolithic−Early Copper Age GHP and among different sites of the Early Copper Age hint at changes in the community organization and potential population contraction or isolation in the Polgár microregion. While severe internal or external social conflict could have caused such phenomena, no traces of violent actions were found at the sites of the Polgár microregion. Furthermore, internal social conflict and the growth of

social inequality might be interrelated. The Copper Age is often characterized by the appearance of institutionalized or hereditary social inequality[15,16]. One of the main arguments supporting this is the correlation between the protein-rich, otherwise distinct diet and the wealth of certain Copper Age burials, e.g. the Varna cemetery[81,82], but the stable isotope analyses do not detect any significant dietary differences among the Early Copper Age individuals on the GHP[83]. The trend of abandoning the Late Neolithic tells and large flat settlements on the GHP can be witnessed on several sites. According to a radiocarbon dating project focused on the Early-Middle Copper Age on the GHP, the establishment of formal cemeteries occurred after 4400−4350 cal BCE[22] (Supplementary Figs. 17-18, Supplementary Data 2). Despite the availability of hundreds of AMS dates for the Late Neolithic and Early Copper Age periods on the GHP, only a few fall within the gap between 4450 and 4350 cal BCE. There are few burials and tiny farmsteads consisting of two or three buildings from this transitional period (e.g. Polgár-Nagy-Kasziba, Rákóczifalva-Bagi föld, site 8, Vésztő-Bikeri)[22,84]. These radiocarbon data corroborate drastic population changes at the turn of the Late Neolithic and Early Copper Age in the Polgár microregion. However, further site-specific investigations are needed to reveal the underlying causes of these demographic and social changes, which are crucial for understanding the broader transformations that took place during this transitional era of the Carpathian Basin.

Our case studies highlight the risks of overgeneralizations in both archaeological and genetic research, where broad conclusions about an entire archaeological culture, region, or period are drawn from the analysis of a single site. Behind the diversity of material culture and cultural traditions, there may have been a range of cultural and social relationships with different dynamics, of which biological relatedness was only one and social ties were another among the bonds that held the community together. The example of the Early Copper Age cemeteries we have presented illustrates that archaeologically similar phenomena can also be underpinned by local communities organized in fundamentally different ways.

## Methods
### Ethics
This research complies with all relevant ethical regulations. While archaeological skeletal remains are not considered human subjects and thus are not covered by Human Subjects regulations, we obtained formal permission for analysis of all samples from local authorities (Satu Mare County Museum, Budapest History Museum, Hungarian Natural History Museum, and Intercisa Museum). Every sample is represented by stewards such as archaeologists or museum curators, who are authors of this paper. All samples studied here were analyzed with the goal of minimizing damage.

### Materials
We sampled eight archaeological sites from present-day Hungary and Romania (see Supplementary Note 1 for full archaeological details).

*Aszód-Papi földek* is a Late Neolithic site in the Gödöllő hillside, Northern Hungary, where among the settlement features, 224 graves were found. The burial activity can be dated from 4760−4700 cal BCE (68.3%) to 4710−4640 cal BCE (68.3%) based on Bayesian-modeled AMS radiocarbon data (Supplementary Fig. 21). The material culture reflects intensive interaction both with the GHP and Transdanubia. Culturally, the Aszód-Papi földek site belongs to the East Transdanubian group of the Lengyel complex, but both Lengyel- and Tisza-style pottery appears at the site[25,85–88].

The *Budapest-Albertfalva-Hunyadi János út* site is situated on the western bank of the Danube River in southern Budapest. Four Early Copper Age features contained human remains with Ludanice-style sherds, stone tools and a grinding stone[89,90] (Supplementary Data 2).

Based on Bayesian-modeled AMS dating, the burials can be dated from 4395−4180 cal BCE (68.3%) to 4160−3930 cal BCE (68.3%) (Supplementary Figs. 22-23, Supplementary Data 2).

The *Iváncsa-Lapos* site is located by the Danube River on a 560−570 m hill range. A single grave (Feature 335/A), dated to the Early Copper Age, was found at the site. It can be dated to 4325−4070 cal BCE (68.3%) (Supplementary Data 2).

The complex site of *Polgár-Csőszhalom* is situated in Northeastern Hungary, in the middle of the so-called Polgár Island in the Upper Tisza Region. It consists of a tell surrounded by a multiple rondel, a smaller double rondel and a large flat settlement[91]. The Late Neolithic site is characterized by the co-occurrence of various pottery styles typical of the Tisza-Herpály-Csőszhalom and Lengyel complexes. The graves, located in various areas of the flat settlement, formed small groups consisting of two to five burials. The majority of the graves were dug in association with houses. Based on Bayesian-modeled AMS radiocarbon measurements, the graves of the single layer settlement can be dated from 4799−4769 cal BCE to 4673−4652 cal BCE (68.3%)[92].

The *Polgár-Nagy-Kasziba* site is situated in Northeastern Hungary, in the so-called Polgár Island in the Upper Tisza Region. Four Early Copper Age graves were found in the northern edge of the excavated area. The graves contained a large number of grave goods, including Tiszapolgár-style pottery[93]. Based on Bayesian-modeled AMS radiocarbon measurements, the graves can be dated from 4520−4360 cal BCE to 4335−4270 cal BCE (68.3%)[22].

The *Rákóczifalva-Bagi föld, Site 8* is situated on the eastern bank of the Tisza river in the Middle Tisza region. Traces of an Early Copper Age settlement (a building, two wells and several pits surrounded by a ditch) and nine contemporary burials were found. Stylistically, the pottery of the graves exhibits characteristics of Tiszapolgár-Kisrétpart-style[22,94]. Based on Bayesian-modeled AMS radiocarbon measurements, the site can be dated from 4525−4380 cal BCE to 4335−4225 cal BCE (68.3%)[22].

The *Tiszapolgár-Basatanya* site is situated in Northeastern Hungary, in the so-called Polgár Island in the Upper Tisza Region. One Late Neolithic and 154 Early Copper Age graves were excavated here in the 1950s, which are considered to present the complete Copper Age cemetery. Bognár-Kutzián distinguished period I (Tiszapolgár, Early Copper Age) and period II (Bodrogkeresztúr, Middle Copper Age), and a few graves with transitional character[95], whose chronological periodization has since been reevaluated as overlapping pottery styles in time[22,96]. Based on Bayesian modeled AMS radiocarbon measurements, the cemetery can be dated from 4355−4270 cal BC to 4150−3990 cal BCE (68.3%), with a 135−305 year (68.3%) span of use (Supplementary Figs. 17-18, Supplementary Data 2).

The *Urziceni−Vamă* Early Copper Age cemetery, with three Neolithic and 129 Copper Age graves, is located in the border zone between Romania and Hungary. Based on Bayesian-modeled AMS radiocarbon measurements, the Copper Age occupation of the site can be dated from 4285−4130 cal BCE to 4035−3940 cal BCE (68.3%) with a 140−300 year (68.3%) span of use (Supplementary Figs. 19-20, Supplementary Data 2). The graves contained Bodrogkeresztúr and Bodrogkeresztúr-Salcuța-style pottery[97–102].

### Radiocarbon dating
AMS radiocarbon data were calibrated using the IntCal20 curve[103] and the OxCal (v4.4.4) software[104]. Site-based Bayesian models were built taking vertical stratigraphy into consideration (Supplementary Figs. 17-23). Bayesian modeled AMS dates (68.3% probability) were used for dating certain graves and sites. Codes of OxCal modeling can be found at GitHub (github.com/ArchGenIn/Szecsenyi-Nagy_2025) and at Zenodo (DOI: 10.5281/zenodo.15221967).

## Sampling

The individuals studied here were all analyzed with the goal of minimizing damage, with permission from local authorities in each location from which they came. Every sample is represented by stewards such as archaeologists or museum curators, who are either authors of this paper or named in the Acknowledgments.

In the case of the Budapest-Albertfalva-Hunyadi János út, Iváncsa-Lapos, Polgár-Nagy-Kasziba, and Urziceni-Vamă sites, we collected samples from all available graves. In the case of Aszód-Papi földek, Polgár-Csőszhalom, and Tiszapolgár-Basatanya, we aimed to sample 30 graves from each site. We selected specific graves for sampling in order to represent both females and males, and burials with both wealthy and modest grave goods, as well as different parts of the burial grounds. Furthermore, in the case of Tiszapolgár-Basatanya, we aimed to sample burials from a grave row to test whether closely related individuals were buried next to each other or whether different principles determined their arrangement. Moreover, we prioritized burials for which previous radiocarbon dating and strontium isotope data had been published to maximize the information and minimize the damage.

## Ancient DNA data generation

A total of 125 samples were prepared at the University of Vienna. Petrous bones were prepared following the method of Pinhasi et al.[105] by extracting the cochlea with a sandblaster, followed by milling. Teeth were cleaned using a dental sandblaster, and selected fragments powdered using a mixer mill. The powders were then shipped to the Harvard laboratory or extracted directly in Vienna, and DNA extracts were shipped to Harvard.

At Harvard Medical School and Vienna University, DNA was extracted from the powders either with the manual procedure using spin columns[106,107] or at Harvard Medical School with the automated ('robotic') procedure using silica magnetic beads and Dabney Binding Buffer on the Agilent Bravo NGS workstation[108]. DNA (from extracts generated in Vienna or Boston) was then converted into barcoded double-stranded partial uracil-treated libraries[109] or USER-treated single-stranded libraries[110] using automation, and then enriched in solution for sequences overlapping 1.24 million SNPs [1240k[111], Twist[112]], as well as for the mitochondrial genome (either spiked into the 1240k or Twist reagent, or in an independent capture). Catalog numbers of the reagents and sequences of oligonucleotides used in this study are listed in Supplementary Data 13.

For each library, we sequenced ~20 million reads of the enriched library (average 19.63 million) using Illumina instruments [Next-Seq500, HiSeq X]; we also sequenced several hundred thousand sequences of the unenriched libraries.

We generated additional captured data for six already published samples (Supplementary Data 1 I-ID_enhanced). For an additional 18 samples, we generated deeper shotgun data (from the same libraries used for capture) with an average genome coverage of 3.8 × using Illumina HiSeq X instruments (Supplementary Data 1).

## Bioinformatic analysis

Samples were sequenced to generate raw paired-end reads; these were prepared for analysis by performing the following steps: preprocessing, alignment, and post-alignment filtering to enable variant calling. Raw reads were demultiplexed using identifying barcodes and indices to assign each read to a particular sample, prior to stripping these identifying tags from the sequences. Paired-end reads were merged into a single molecule using the overlap of the reads as a guide. Single-end reads were aligned to the hg19 human reference genome (https://www.internationalgenome.org/category/grch37/) and the basal Reconstructed Sapiens Reference Sequence (RSRS)[113] mitochondrial genome using the samse aligner of BWA[114]. Duplicate molecules were marked based on barcoding bin, start/stop positions and orientation.

For calling variants, a pseudo-haploid approach was used at targeted SNPs, where a single base is randomly selected from a pool of possible bases at that position, filtering by a minimum mapping quality of 10 and base quality 20, after trimming reads by two base pairs at both the 5′ and 3′ ends to remove damage artifacts. Computational pipelines with specific parameters are publicly available on GitHub at: https://github.com/dReichLab/ADNA-Tools and https://github.com/dReichLab/adna-workflow, and use: BWA (v0.7.15-r1140), MarkDuplicates (v2.17.10) (https://github.com/broadinstitute/picard), haplogrep (v2.1.1), ANGSD (v0.921-3-g40ac3d6), preseq (v2.0.3), pmdtools (v0.60.5).

## Genetic sex determination

We determined molecular sex based on the ratio of Y-chromosome to the sum of X- and Y-chromosome sequences. Ratios <0.03 were interpreted as confidently female, and ratios >0.3 were interpreted as confidently male (Supplementary Data 1).

## X-chromosome contamination in males

X-chromosomal contamination of male individuals was measured with two software packages: ANGSD (v0.939-10-g21ed01c, htslib 1.14-9-ge769401) and hapCon (hapROH package v0.60). The doCounts for the former method was run on X-chromosomal regions from positions 5,500,000 to 154,000,000, with base quality 30 and mapping quality 25. The toolkit's contamination executable was used to process the resulting file. Parameters were set to -b 5500000 -c 154000000 -d 3 -e 100 -p 1, and the file used by the -h parameter was included with the software by the developers.

For the second method, the required mpileup file was created with samtools[115] mpileup (v1.10, htslib 1.10.2-3ubuntu0.1), base quality was set to 30, and mapping quality was set to 25. Parameters were used according to the official documentation. All necessary reference files used were included by the developers in the software package.

## Mitochondrial DNA analyses

Mitochondrial haplogroups were determined using Haplogrep[116] v2.1.25, which utilizes Phylotree mitochondrial DNA tree[117] Build 17. We estimated mitochondrial contamination using contamMix[118].

We aligned sequences in each group with ClustalO within Ugene v40[119]. The alignments were checked and corrected manually where necessary. Compared to the rCRS (revised Cambridge Reference Sequence), we deleted the following ambiguous positions: np 42, 57, 291–317, 447–458, 511–524, 568–573, 594–597, 1,718, 2,217–2,226, 3,106–3,110, 3,159–3,167, 5,890–5,894, 8,272–8,281, and 16,184–16,193. Median Joining network (nexus) were created with DnaSP v5[120] and the figures were drawn with the PopArt v1.7 (Population Analysis with Reticulate Trees) program[121].

## Compilation of data

Genotypes of 125 + 6 individuals were merged with previously published genotypes of ancient and modern individuals obtained through 1240k capture or shotgun sequencing, downloaded from the Allen Ancient DNA Resource (AADR)[122]. Especially important Neolithic-Copper Age data were obtained from selected literature[4–7,58,123].

## Principal component analysis (PCA)

We used the Eigensoft (v8.0.0) package to perform principal component analysis with Smartpca software[56]. Specifically, we projected the genotypes of the ancient individuals onto a West-Eurasian basemap of genetic variation using shrinkmode, which we calculated using the HO panel of the AADR[122]. PCA plots were constructed in Python 3.12 using the matplotlib package[124] (v3.9.2). To detect outliers within the main analysed groups, based on PC1 and PC2 values, we used the Mahalanobis Distance approach[125], which calculates the distance of each data point from the group mean while accounting for the correlation

 

between variables. First, we computed the mean vector for the group across PC1 and PC2, followed by the calculation of the covariance matrix for PC1-2. The Mahalanobis Distance for each individual was then calculated, considering both the deviation from the group mean and the covariance matrix. To assess statistical significance, we calculated the $p$-value for each individual's Mahalanobis Distance using the chi-squared distribution with two degrees of freedom, using Scipy[126]. If the $p$-value was below the specified threshold ($p = 0.05$), the individual was flagged as a potential outlier (Supplementary Data 3). We used Levene's test and Bartlett's test to assess the homogeneity of variance between groups or sites, based on PC1-2. We used both tests, to cover both normally and non-normally distributed datasets adequately and to check for equality of variances.

## $f_4$-statistics

We used AdmixTools (v7.0.1)[59] for $f_4$-statistics, in the forms of $f_4$(Israel_Natufian, Turkey_Epipalaeolithic, Test, Mbuti), $f_4$(Russia_EHG, Loschbour, Test, Mbuti), and $f_4$(CHG, Russia_EHG, Test, Mbuti), in order to test allele sharing differences between the test individuals and the given references. $f_4$-scores are expected to follow a normal distribution (Supplementary Fig. 2), and outliers were detected with Z-score (Supplementary Data 4).

## qpAdm modeling

Several 1-4-way qpAdm models[60,61] were constructed to characterize the study groups and individuals, to test for local continuity on the GHP, and to test for potential streams of migration into the region leading to ancestry outliers. Outgroups and source populations are listed in Supplementary Data 5. We used the following HG sources: WHG (including, as proxies, Mesolithic genomes from England and sites such as Loschbour and Bad Dürrenberg) and EHG (including Mesolithic genomes from Ukraine and Russian Karelia), along with CHG (Caucasus hunter-gatherer) to trace potential influxes and variability. Variable HG proportions have previously been shown to provide a powerful tool for distinguishing farmer communities. The intermediate WHG-EHG genetic composition of the Serbian Iron Gates Mesolithic population has been described previously, anticipating the intermediate or dual composition of the local HG sources in the Carpathian Basin as well[5,127]. We excluded low-coverage individuals (<50k SNP on the 1.15 million autosomal targeted SNPs) from these analyses. In the group-based qpAdm, we excluded one member of each first-degree relative pair. A two-sample $t$-test was used to test the null hypothesis that there is no significant difference (at the significance level of 0.05) in a given component between pairs of selected populations. We computed the Z-score using the Pandas v2.2.2 library in Python 3.12, to test how far an individual's EHG or WHG component value deviates from the group mean for groups with $n > 3$, assuming that the WHG data are normally distributed and the EHG components follow half-normal or truncated normal distribution (Supplementary Fig. 2). Z-scores beyond ±2 indicate that the individual's value is significantly different from the group mean (in terms of standard deviations) (Supplementary Data 5).

## Determination of Y-chromosome haplogroups

Sequencing data (BAM files) were screened with Yleaf[128] (v2.2) for SNPs based on ISOGG Y-chromosomal tree v17 (https://isogg.org/tree/).

## Kinship analysis—KIN, READ

To assess biological relatedness, the combination of three methods were used: READ (v1.01), KIN (v3.1.1), and ancIBD (v0.5), as described later.

The READ analysis[49] can identify biologically related individuals up to the second degree based on the proportion of non-matching alleles, providing valuable information about close biological relatedness.

The KIN analysis[50] estimates the relatedness of a pair of individuals by analyzing the identical-by-descent segments they share and can accurately classify up to third-degree relatives. It can differentiate between siblings and parent-child pairs.

## Genotype imputation

For imputation, we applied the GLIMPSE (v1.1.1)[129] software with the 1000 Genome Project as the reference panel on VCF files to estimate genotype posterior at bi-allelic SNP sites. We restricted the IBD analyses to SNPs in the 1240k capture, which are informative for ancient DNA studies. These VCF files were generated using bcftools mpileup (v1.10.2)[130] applied on sequence data in aligned BAM format. A full description of the imputation pipeline is provided in Ringbauer et al.[51].

## Identity-by-descent chromosome segment sharing

The ancIBD program identifies IBD segments longer than eight centimorgan (cM) in aDNA data. This method can distinguish up to sixth-degree relatives which are expected to share multiple long IBD segments and which provide strong signals for identifying close and distant biological relatives[51]. Before calling IBD with ancIBD (v0.5), samples were filtered based on coverage (at least 600k SNPs hit on autosomal targets). Following filtering, the analysis included: a total of 26 samples from Polgár-Csőszhalom, 7 from Aszód-Papi földek, 29 from Tiszapolgár-Basatanya, 63 from Urziceni-Vamă, 4 from Budapest-Albertfalva-Hunyadi János út, and one sample each from Rákóczifalva-Bagi föld site 8, Polgár-Nagy-Kasziba and Iváncsa-Lapos. We performed an ancIBD run with high quality Late Neolithic-Copper Age genomes, adding samples from previously published articles, in order to observe the genetic connectivity with other European Neolithic and Copper Age populations[4,5,7,8,52,57,63,64,111,123,131–139].

IBD segment sharing was visualized with Gephi v0.10.1[140] using different layout algorithms: ForceAtlas2[65] with 0.04 gravity and scaling=10 and manually constructed layout, which corresponds to the map of the studied sites. The Leiden algorithm was applied in Gephi with the constant Potts model with 0.04 resolution and 10,000 iterations[66].

After filtering for edges >12 cM, 269 samples remained in the Late Neolithic-Copper Age graph. We calculated the degree of centrality ($k$) of a node counting links connected to all other nodes in the network. This degree of centrality was normalized: each node was divided by the maximum possible number of connections it could have (which, in an undirected graph, is n-1, where n is the number of nodes in the graph).

Maximum IBD segment lengths were used as weights of the links attached to each node (w). These weights were analysed in two categories: "within strength" for the node represents the sum of edge weights connecting it to other nodes within the same module, whereas "between strength" for the node represents the sum of edge weights connecting it to nodes in other modules, considering the archaeological site of the burial as a module.

The clustering coefficient (C) for each node in a graph is a measure of how interconnected a node's neighbors are. We used the degree_centrality and nx.clustering function of NetworkX[141] (v2.3) and Gephi (v0.10.1) for defining C and $k$, and plotted with seaborn[142] v0.9.0. Statistical tests implemented in SciPy v1.7.3 for the differences and similarities of these values were Welch's $t$-test and Fisher's exact test for $k$ and the two-sample Kolmogorov-Smirnov tests for the distribution of the individual data ($k$ and C) by sex[126].

We counted cliques (complete subgraphs) with minimum node size of 3 within the whole graph (created in NetworkX), differentiating between and within-module cliques and sexes. A within-site clique is a clique where all nodes belong to the same site. A between-site clique is a clique where nodes belong to more than one site. We performed the Chi-square test for each site and sex group separately, using expected

frequencies (Supplementary Data 7B). Scripts for statistical calculations and plotting can be found at GitHub (github.com/ArchGenIn/Szecsenyi-Nagy_2025) and at Zenodo (DOI: 10.5281/zenodo.15221966).

IBD segment sharing within and among the populations was also used to estimate differences in diploid effective population sizes of the communities, using the TTNe Python package v0.0.1a0[143]. Filtering was applied for non-related individuals (12 < 100 cM segments shared). The 95% confidence interval (CI) of $N_e$ was calculated via maximum likelihood approach (as implemented in the TTNe algorithm). Results were plotted with the same program package (TTNe.analytic).

### Runs of homozygosity analyses

The hapROH software[62] can detect signals of recent inbreeding or indicate a small effective population size. The hapROH v1.0 program was run with default parameters for all pseudo-haploid genotypes with at least 300k SNP covered. Samples with a minimum of 400k SNP were used in the further analyses. Plotting was conducted using the multiplot function of this program. The $N_e$ vignette of this program was used to estimate effective population sizes with confidence intervals. For the $N_e$ calculations, we removed individuals with >50 cM sum ROH above 20 cM stretches. Statistical inferences were made using Fisher's exact test of the scipy.stats module and permutation test (with 10,000 permutations) of the SciPy[126] v1.7.3 and Numpy[144] v1.21.6 libraries in Python 3.7.

### Reporting summary

Further information on research design is available in the Nature Portfolio Reporting Summary linked to this article.

## Data availability

All data needed to evaluate the results of the paper are present in the paper and/or the Supplementary Materials. Ancient genome sequences were uploaded to ENA (European Nucleotide Archive), under the accession number PRJEB86386. The AADR v54.1 dataset is publicly available at https://dataverse.harvard.edu/dataset.xhtml?persistentId=doi:10.7910/DVN/FFIDCW. 1000 Genome Project data, used as the reference panel for the imputation was taken from https://www.internationalgenome.org/data-portal/data-collection/30x-grch38. Data required to generate all figures in the manuscript are available in Supplementary Data files and Source Data files. Open science principles require making all data used to support the conclusions of a study maximally available, and we support these principles here by making fully publicly available not only the digital copies of molecules (the uploaded sequences) but also the molecular copies (the ancient DNA libraries themselves, which constitute molecular data storage). Researchers who wish to carry out deeper sequencing of libraries published in this study should make a request to corresponding author, D.R. We commit to granting requests as long as the libraries remain preserved in our laboratories, with no requirement that we be included as collaborators or co-authors on any resulting publications. Source data are provided with this paper.

## Code availability

Specific new codes used for satistical testing and plotting were uploaded to GitHub [github.com/ArchGenIn/Szecsenyi-Nagy_2025] and Zenodo [https://doi.org/10.5281/zenodo.15221967].

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

## Acknowledgements

We thank the technical support and laboratory assistance provided by the following people from Harvard Medical School and the Vienna University: Nicole Adamski, Nasreen Broomandkhoshbacht, Kim Callan, Elizabeth Curtis, Ann Marie Lawson, Megan Michel, Susann Nordenfelt, Jonas Oppenheimer, Lijun Qiu, Kristin Stewardson, Noah Workman, Fatma Zalzala, Kirsten Mandl, Kadir Toykan Ozdogan, Sarah Kellie Duffett Carlson, Beatriz Gamarra Rubio, Anna Wagner, Lea Demetz, Stefanie Hofer and Guillermo Bravo. We thank Noémi Borbély for her support in the uniparental data analyses and Balázs G. Mende for his administrative support. This study was funded by the MTA-ELTE Lendület "Momentum" program of the Hungarian Academy of Sciences. T.H. was supported by the Bolyai Scholarship of the Hungarian Academy of Sciences. The ancient DNA data generation and analysis at Harvard was supported by the National Institutes of Health (R01-HG012287), the John Templeton Foundation (grant 61220), by a private gift from Jean-Francois Clin, by the Allen Discovery Center program, a Paul G. Allen Frontiers Group advised program of the Paul G. Allen Family Foundation and by the Howard Hughes Medical Institute (DR).

## Author contributions

Conceptualization: A.An., P.R., R.P., D.R., Zs.S., Data Collection: C.V., A.An., P.R., T.H., K.K., T.Sz., S.É., T.K., Zs.V., O.Ch., R.P., Zs.S., Data Curation: K.J., S.M., Zs.S. Formal Analysis: A.Sz-N., H.R., S.M., A.Ak., Zs.S. Investigation: A.Sz-N., K.J., N.R., O.Ch., Zs.S. Writing – Original Draft: A.Sz-N., Zs.S. Writing – Review & Editing: A.Sz-N., H.R., P.R., D.R., Zs.S., Resources: D.R., Zs.S. Software: K.J., H.R., S.M., A.Ak.

## Funding

## Competing interests

The authors declare no competing interests.
