## [Transparent Peer Review file · Nature Communications]

Ancient DNA reveals diverse community organizations in the 5th millennium BCE Carpathian Basin

Corresponding Author: Dr Anna Szécsényi-Nagy

Version 0:

Reviewer comments:

Reviewer #1

(Remarks to the Author)

Commenting from the perspective of an archaeologist, I have very little to add. The study presents very exciting insights into the social dynamics of Late Neolithic / Copper Age people in the Carpathian Basin. Not only does it demonstrate a genetic connection between LN/CA populations, it also shows the diversity of CA populations and communities' cemetery organization.

The Basatanya case study is particularly enlightening. It would be interesting to discuss which arguments speak for this being a biological or social effect, i.e. if the genetic patterns seen are more likely to be an effect of a shrinking and isolating population or indeed a sign of social hierarchization, with elites pursuing a particular strategy to preserve their lineage. In addition to burial wealth, health indicators, body height, spatial data etc. may be utilized.

It is of course desirable to sample all individuals of a site for a full picture of the population in the future, but it is clear that there are resource constraints.

To find Family B among all the burials seems a lucky strike – was there any indication of kinship connections in the archaeological context before the individuals were sampled?

Pottery styles are mentioned as cultural markers of social groups, but not further explained. It would be great to have a quick definition of what the styles entail. No explanation is given of why these may have varied, apart from potential ethnic entities.

For readability, there are too many abbreviations used. It would be helpful to use abbreviations only for chronological and genetic concepts, not for geographical entities (write out GHP and CB).

In line 224, CHG appears. I do not think this is defined earlier in the paper or in the abbreviations list. Central Hunter-Gatherers?

C14 dates: I do not understand why the calibrated dates are most often, but not always given as range to range dates (e.g. 4760-4700 cal BCE (68.3%) to 4710-4640 cal BCE (68.3%), why not 4760-4640 cal BCE (68.3%). What does this extra range in range add?

line 765: pottery, not potteries

line 995 and 996: We thank... for her support.... for his....

line 1291: Frühe Kulturen aus Rumänien

supp line 78 etc: do not use crouched – they are flexed burials (see Knüsel 2014 Crouching in fear: Terms of engagement for funerary remains)

supp 176:were male... (it is better to speak of sex estimation, not determination: 27.88% of the burials were estimated as males,)

Reviewer #2

(Remarks to the Author)

- What are the noteworthy results?

There are actually many results that are interesting from a period-specific point of view, such as the possibility that there was increasing demographic isolation, perhaps population decline, between communities in the CA compared to the LN. As this is not connected to these communities dropping out of cultural innovation networks, it will give scholars plenty to think about. The most important result, however, is the crucial role of variability within even the same region and time period. This is a conclusion that needs to be taken much more seriously across the aDNA field in prehistoric Europe, and indeed beyond, and I hope many people not only read it, but also really take it to heart in their analysis. This alone would make the paper worth publishing, but as I said, it also has many, many other important results at a regional level.

- Will the work be of significance to the field and related fields? How does it compare to the established literature? If the work is not original, please provide relevant references.

Yes, see above. I cannot answer to whether the methods are in any way novel, but it will be of significance for prehistoric archaeology, where there is an increasing trend to want to compare kinship structures, thanks also to recent grants. The paper provides a very thoughtful exploration of these topics, which makes it stand out positively compared to some of the brasher, but also less reliable "headline-focused" papers who claim to have found "the" solution. This paper is exemplary in pointing out where different possibilities exist, and in providing coherent interpretations without losing sight of the alternative possibilities. It is also a very sound approach to not start with "culture" as the main block for analysis.

- Does the work support the conclusions and claims, or is additional evidence needed?

In general yes. But as a rather traditional archaeologist with certain scientific limitations, I would like to have clarification on a few issues. These are evidently quite logical to the authors, but I need a bit more of a walk-through. In detail, these are:

- Small things: In line 85, use "central Europe" instead of "Europe", as there are many areas of Europe (Spain, Greece...) for whom the CB does not play a large role in Neolithisation. In line 90, I suggest to delete "human" at the start – maybe but NL/ECA, but it gave me an association of global relevance, when things like metallurgy developed in several world regions.
- Are the two male outliers at Basatanya different in terms of grave goods from the remainder of the community? It is worth stating this explicitly.
- Lines 437/438 should this be "predominantly patrilocal and patrilineal", seeing as we also have relatives connected through the maternal line only, plus grown-up daughters?
- Lines 505-506: I can't follow the logic of how the pedigree informs on pottery chronology here. Can you make this clearer?
- In table 1, in the last row – can you distinguish in the "non patrilineal" whether matrilineal or bilateral would be more likely? If not, maybe make explicit that you have considered different possibilities, but that those would all look similar, or indeed that it does not correspond easily to any of the usual patterns. Or are you cautious because most of the outliers here still are women, and this is normally interpreted as patrilocality/patriliny?
- The CA population shares a lot of maternal lines with the LN (line 546). So what is happening to the men, especially given that some CA sites at least are patrilineal. Is this a change in practice at this transition?
- Paragraph starting line 625: is it worth pointing to the possibility that wealth inheritance may not be the only reason for first-cousin marriage after all? If you look at all these large-scale comparative studies, they only ever give trends and likelihoods, it's never the answer for all of the communities; this can be mentioned quite directly.
- The complete absence of fathers at Urziceni is interesting, and should be interpreted in some way. Were they returned to their natal communities for burial in a matrilineal (but not matrilineal) system? See Bradley Ensor on post-mortem mobility
- 672-674 The jump from grave goods to isotopes as a measure of wealth is potentially a little confusing to the more casual reader. Presumably, you mean that higher protein should be connected to higher social status and therefore should correlate with more/nicer grave goods, but it works better to spell this out.

- Are there any flaws in the data analysis, interpretation and conclusions? - Do these prohibit publication or require revision?

Not that I can see. I think there is one aspect that is a bit of an omission. Towards the end, the authors mention (in lines 670-ish and onwards) the question of inheritance of wealth. This is a big question for the CA, but has not really been explored in the main data-focused parts of this paper. Of course, it would add a lot of words, so the authors may well be planning another article on this. Even in this case, however, it would be worth mentioning whether a wealth analysis of grave goods in the cemeteries sampled here has been/is being carried out, and what it seems to indicate (with a hint that full details are to come, if that is indeed the case). Is there hierarchy based on "grave wealth" at the sampled cemeteries, and does it show potential inheritance within family groups? Varna is, after all, quite over the top and almost unique (together with Durankulak) in the range of difference, so it's worth to stress any variability here also.

- Is the methodology sound? Does the work meet the expected standards in your field?

I can't really comment on this aspect, but the Excel table with the archaeological detail answers well to the needs of prehistorians. For the written supplementaries, I only checked the parts pertaining to archaeological sites. These are fine, though I question the point of giving two percentage points (e.g. 25.93%, as opposed to 25.9, or indeed even just 26% for Polgar children) both here, and then also in the main text and tables – it suggests a level of accuracy that is a little meaningless in this context and given the absolute numbers involved, especially since a much more pragmatic policy has been adopted where 14C likelihoods are quoted.

In this sentence: "Wealthy burials were furnished with limestone beads – as belts in the case of women –, copper ornaments and Volhynian flint blades and stone mace heads in the case of men" it is not quite clear where the female grave goods list ends and the male one begins. Just make two sentences for clarity. It would also make sense to add at the start of the supplementary section that further details on the graves sampled are included in the Excel sheet, just to ensure the readers find it.

In general, the level of English is very good, but I would suggest professional proof-reading, as a few things ("children

graves" instead of "child greaves", the odd issue with definite articles, having been versus have been) have slipped through, in both supplementaries and main text.

- Is there enough detail provided in the methods for the work to be reproduced?

Again, I can't comment. I also did not read the "methods" section in the PDF beyond page 26 (apart from the ethics statement and the bit about sampling choices, which are fine).

Reviewer #3

(Remarks to the Author)

Szécsényi-Nagy et al. analysed SNP capture data from 125 individuals from the Carpathian Basin, dating from the Late Neolithic to the Early Chalcolithic. They applied commonly used population genetics tools, kinship software, and IBD analysis to investigate social organization, genetic continuity, and admixture patterns across different sites in present-day Hungary and Romania.

The authors claim that, despite differences in cultural epochs and burial practices, they found a relatively high degree of genetic continuity in the Carpathian Basin during this period. Another key finding is the lack of correlation between material culture and kinship organization systems. Two cemeteries that are close in time, space, and material culture exhibit different patterns of consanguinity, as well as variations in genetic diversity and social organization.

This study provides new insights into population transitions and human movements from the Late Neolithic to the Early Chalcolithic in the Carpathian Basin. It also contributes to our understanding of social organization during this period. The conclusions are well supported by the results, and the methodology is replicable. I appreciate how the authors support all their observations with statistical tests, something that is sometimes lacking in archaeogenetic studies. I recommend acceptance with minor changes.

Minor Comments and Suggestions

The authors conducted three contamination estimates: two based on the X chromosome (valid only for males) and one using contamMix for mitochondrial DNA. However, for the latter, only some samples have results listed in the "Assessment Note" column. I guess that the remaining samples have contamMix results close to zero. If this is the case, could the authors include a note in the Supplementary Information (SI) specifying that only samples with contamMix results above a certain threshold are annotated?

PCA: I acknowledge that due to the high number of samples, it is difficult to display a clear figure. I find myself having trouble interpreting the PCA. Perhaps highlighting the newly reported samples or adding a zoom panel that focuses solely on the new samples, without the references, would help clarify the presentation.

Why did you choose to use only PC1 and PC2 for detecting outliers in the main analysed groups? Would incorporating the first 10 PCs provide a more robust detection?

Lines 175 and 179: Be cautious when inferring ancestry and genetic affinities based solely on PCA positions. While PCA provides useful insights into genetic relationships, it does not directly quantify ancestry or capture the complexity of population interactions. Therefore, drawing firm conclusions about genetic affinities based solely on PCA positions may oversimplify the results. I suggest rephrasing these sentences.

There is a discrepancy in the number of PCA outliers reported. In the results (lines 186–187) and Table 1 for Urziceni-Vamă, the ECA states that there are 8 PCA outliers. However, in the discussion (line 555), it mentions 11 outliers, 8 of whom are female. Should the correct figure be 8 outliers, with 5 of them being female? Please confirm.

To assess genetic continuity and varying degrees of admixture, the authors rely on f_4 -statistics, and qpAdm analyses, which are well-established methods for detecting admixture between populations. However, qpAdm results can vary significantly depending on the choice of reference populations, often producing plausible models with alternative left populations beyond those selected by the authors. Can the authors provide more detailed justification for their selection of reference (right) populations for each case and explain how their results might change if different but similar populations were used instead?

In Figure 2B, could you clarify in the figure caption what the black error bar represents?

In the qpAdm modeling, the authors excluded one individual from each first-degree relative pair. However, shouldn't second- and even third-degree relatives also be excluded from this analysis?

Version 1:

Reviewer comments:

Reviewer #1

(Remarks to the Author)

Thanks for the revised version, I have no more comments.

Reviewer #2

(Remarks to the Author)

Thank you for taking all the suggestions so seriously - I really hope that this excellent paper will get the impact it deserves, and I shall certainly be advertising it once it's out! Looking forward to your follow-up studies as well.

Reviewer #3

(Remarks to the Author)

I support publication and have no further comments.

This is the point-by-point answer to the reviews, corresponding to the manuscript Szécsényi-Nagy et al.:

REVIEWER COMMENTS

Reviewer #1 (Remarks to the Author):

Commenting from the perspective of an archaeologist, I have very little to add. The study presents very exciting insights into the social dynamics of Late Neolithic / Copper Age people in the Carpathian Basin. Not only does it demonstrate a genetic connection between LN/CA populations, it also shows the diversity of CA populations and communities' cemetery organization.

The Basatanya case study is particularly enlightening. It would be interesting to discuss which arguments speak for this being a biological or social effect, i.e. if the genetic patterns seen are more likely to be an effect of a shrinking and isolating population or indeed a sign of social hierarchization, with elites pursuing a particular strategy to preserve their lineage. In addition to burial wealth, health indicators, body height, spatial data etc. may be utilized.

It is of course desirable to sample all individuals of a site for a full picture of the population in the future, but it is clear that there are resource constraints.

To find Family B among all the burials seems a lucky strike – was there any indication of kinship connections in the archaeological context before the individuals were sampled?

The general sampling strategy is described in the paper's Method section, and we added two sentences about the sampling strategy for the Tiszapolgár-Basatanya cemetery. One of our research questions regarding this site was who was buried next to whom in a row and what the cemetery's organizing principle was. Therefore, we selected samples from different rows and parts of the cemetery.

Reviewer 1 correctly pointed out that financial constraints prevented us from sampling the complete sites. Anna Szécsényi-Nagy and Zsuzsanna Siklósi are currently working on aDNA testing of the complete site, combining this with full archaeological re-evaluation and a new osteoarchaeological examination. Therefore, the issues of transgenerational transmission of wealth or population isolation will be discussed and answered based on the cemetery's complete and detailed multidisciplinary analysis. This thorough evaluation requires another paper. However, we believe this paper already contains significant new results worthy of publication.

The osteoarchaeological examination was performed in the 1950s and early 1960s and was limited to sex and age estimations included in the cemetery's monograph (Bognár-Kutzián 1963). No further information is currently available (pathology, trauma, body height).

Pottery styles are mentioned as cultural markers of social groups, but not further explained. It

would be great to have a quick definition of what the styles entail. No explanation is given of why these may have varied, apart from potential ethnic entities.

We added a short description of pottery styles to the supplementary material with a figure (Supplementary Note 2, Supplementary Fig. 1).

For readability, there are too many abbreviations used. It would be helpful to use abbreviations only for chronological and genetic concepts, not for geographical entities (write out GHP and CB).

We have written out "CB" as "Carpathian Basin" in the text, upon your suggestion. "GHP" stands for "Great Hungarian Plain", and it appears more than 30 times in the text. Therefore, we find it reasonable to use it throughout the text. We reconsidered the other abbreviations as well, and kept only those that were inevitable.

In line 224, CHG appears. I do not think this is defined earlier in the paper or in the abbreviations list. Central Hunter-Gatherers?

Thank you for pointing this out, by now we have complemented the list of abbreviations and also wrote at first mention that CHG stands for Caucasus Hunter-Gatherers.

C14 dates: I do not understand why the calibrated dates are most often, but not always given as range to range dates (e.g. 4760-4700 cal BCE (68.3%) to 4710-4640 cal BCE (68.3%), why not 4760-4640 cal BCE (68.3%). What does this extra range in range add?

These dates refer to Bayesian-modelled start and end boundaries with their own probability distributions. That's why we use ranges.

line 765: pottery, not potteries

Corrected.

kine 995 and 996: We thank... for her support.... for his....

Corrected.

line 1291: Frühe Kulturen aus Rumänien

Corrected.

supp line 78 etc: do not use crouched – they are flexed burials (see Knüsel 2014 Crouching in fear: Terms of engagement for funerary remains)

Corrected.

supp 176:were male... (it is better to speak of sex estimation, not determination: 27.88% of the burials were estimated as males,)

Corrected.

Reviewer #2 (Remarks to the Author):

- What are the noteworthy results?

There are actually many results that are interesting from a period-specific point of view, such as the possibility that there was increasing demographic isolation, perhaps population decline, between communities in the CA compared to the LN. As this is not connected to these communities dropping out of cultural innovation networks, it will give scholars plenty to think about. The most important result, however, is the crucial role of variability within even the same region and time period. This is a conclusion that needs to be taken much more seriously across the aDNA field in prehistoric Europe, and indeed beyond, and I hope many people not only read it, but also really take it to heart in their analysis. This alone would make the paper worth publishing, but as I said, it also has many, many other important results at a regional level.

We sincerely thank the reviewer for appreciating our findings; we also hope they will have an impact on the archaeogenetic community.

- Will the work be of significance to the field and related fields? How does it compare to the established literature? If the work is not original, please provide relevant references.

Yes, see above. I cannot answer to whether the methods are in any way novel, but it will be of significance for prehistoric archaeology, where there is an increasing trend to want to compare kinship structures, thanks also to recent grants. The paper provides a very thoughtful exploration of these topics, which makes it stand out positively compared to some of the brasher, but also less reliable “headline-focused” papers who claim to have found “the” solution. This paper is exemplary in pointing out where different possibilities exist, and in providing coherent interpretations without losing sight of the alternative possibilities. It is also a very sound approach to not start with “culture” as the main block for analysis.

Thank you very much for this comment as well. Indeed, our goal was to thoroughly explore the topic and carefully consider any a priori formation of analytical groups.

- Does the work support the conclusions and claims, or is additional evidence needed?

In general yes. But as a rather traditional archaeologist with certain scientific limitations, I would like to have clarification on a few issues. These are evidently quite logical to the authors, but I need a bit more of a walk-through. In detail, these are:

- Small things: In line 85, use "central Europe" instead of “Europe”, as there are many areas of Europe (Spain, Greece...) for whom the CB does not play a large role in Neolithisation.

Corrected (line 99 in the clean document).

In line 90, I suggest to delete “human” at the start – maybe but NL/ECA, but it gave me an association of global relevance, when things like metallurgy developed in several world regions.

Corrected.

• Are the two male outliers at Basatanya different in terms of grave goods from the remainder of the community? It is worth stating this explicitly.

No, they did not differ from the others. We have added this information to the Results and Discussion, in line 413 (in the clean document).

• Lines 437/438 should this be “predominantly patrilocal and patrilineal”, seeing as we also have relatives connected through the maternal line only, plus grown-up daughters?

Thank you for your remark, we have added it.

• Lines 505-506: I can't follow the logic of how the pedigree informs on pottery chronology here. Can you make this clearer?

Here, we found that Bodrogkeresztúr-Salcuța pottery was only in the graves of the younger generation, and was absent from both generations together as well as from the older generation alone. Therefore, we suggest that this mixed style might appear quickly and restricted only to the younger generation.

• In table 1, in the last row – can you distinguish in the “non patrilineal” whether matrilineal or bilateral would be more likely? If not, maybe make explicit that you have considered different possibilities, but that those would all look similar, or indeed that it does not correspond easily to any of the usual patterns. Or are you cautious because most of the outliers here still are women, and this is normally interpreted as patrilocality/patrilineality?

This is an excellent and challenging question. We considered all of the possibilities listed by Ensor 2021 (see also below), and finally, we decided to be cautious, as strontium isotope data are currently unavailable from the Urziceni-Vamă cemetery. Combining the aDNA with strontium isotope data might help us distinguish between matrilocality, bilocality, and neolocality. Otherwise, the mtDNA haplotype variability is also significant, which speaks against matrilineality at Urziceni.

• The CA population shares a lot of maternal lines with the LN (line 546). So what is happening to the men, especially given that some CA sites at least are patrilineal. Is this a change in practice at this transition?

No, we do not think so. As we could sample 30-30 burials from Polgár-Csőszhalom and Tiszapolgár-Basatanya, there is still the possibility that paternal lines also connected them. Both sites were rather patrilineal, and the connection between them is not directly

linear following a paternal line, but e.g. some women left the Csószhalom community and married men (either in Csószhalom and they are not sampled or at another contemporaneous site in the microregion and not sampled), and their kindreds were found in Tiszapolgár-Basatanya.

• Paragraph starting line 625: is it worth pointing to the possibility that wealth inheritance may not be the only reason for first-cousin marriage after all? If you look at all these large-scale comparative studies, they only ever give trends and likelihoods, it's never the answer for all of the communities; this can be mentioned quite directly.

Corrected.

• The complete absence of fathers at Urziceni is interesting, and should be interpreted in some way. Were they returned to their natal communities for burial in a matrilocal (but not matrilineal) system? See Bradley Ensor on post-mortem mobility

This is another excellent and challenging question. As we have already mentioned, we do not currently have strontium isotope data that would be necessary to discuss the different residence patterns and exclude some of them. Thank you for your suggestion. We consider this a potential explanation and have added it to the text, in lines 676-678 in the clean document.

• 672-674 The jump from grave goods to isotopes as a measure of wealth is potentially a little confusing to the more casual reader. Presumably, you mean that higher protein should be connected to higher social status and therefore should correlate with more/nicer grave goods, but it works better to spell this out.

Corrected.

- Are there any flaws in the data analysis, interpretation and conclusions? - Do these prohibit publication or require revision?

Not that I can see. I think there is one aspect that is a bit of an omission. Towards the end, the authors mention (in lines 670-ish and onwards) the question of inheritance of wealth. This is a big question for the CA, but has not really been explored in the main data-focused parts of this paper. Of course, it would add a lot of words, so the authors may well be planning another article on this. Even in this case, however, it would be worth mentioning whether a wealth analysis of grave goods in the cemeteries sampled here has been/is being carried out, and what it seems to indicate (with a hint that full details are to come, if that is indeed the case). Is there hierarchy based on "grave wealth" at the sampled cemeteries, and does it show potential inheritance within family groups? Varna is, after all, quite over the top and almost unique (together with Durankulak) in the range of difference, so it's worth to stress any variability here also.

As we wrote in the discussion, we compared the biological relatedness and wealth of burials. Reviewer 2 correctly emphasised that complete analyses of the cemeteries are necessary to answer this question properly. This is in progress, and we will publish them in detail in another paper, as we mentioned above and wrote in the text. However, the current sample sizes prevented us from detecting a statistically significant correlation and drawing a firm conclusion about this question.

- Is the methodology sound? Does the work meet the expected standards in your field?
I can't really comment on this aspect, but the Excel table with the archaeological detail answers well to the needs of prehistorians. For the written supplementaries, I only checked the parts pertaining to archaeological sites. These are fine, though I question the point of giving two percentage points (e.g. 25.93%, as opposed to 25.9, or indeed even just 26% for Polgar children) both here, and then also in the main text and tables – it suggests a level of accuracy that is a little meaningless in this context and given the absolute numbers involved, especially since a much more pragmatic policy has been adopted where 14C likelihoods are quoted.

We have rounded these percentage values to one decimal place. Thank you for pointing this out.

In this sentence: “Wealthy burials were furnished with limestone beads – as belts in the case of women –, copper ornaments and Volhynian flint blades and stone mace heads in the case of men” it is not quite clear where the female grave goods list ends and the male one begins. Just make two sentences for clarity. It would also make sense to add at the start of the supplementary section that further details on the graves sampled are included in the Excel sheet, just to ensure the readers find it.

Corrected.

In general, the level of English is very good, but I would suggest professional proof-reading, as a few things (“children graves” instead of “child greaves”, the odd issue with definite articles, having been versus have been) have slipped through, in both supplementaries and main text.

A native-speaker colleague has read and corrected the whole manuscript.

- Is there enough detail provided in the methods for the work to be reproduced?
Again, I can't comment. I also did not read the “methods” section in the PDF beyond page 26 (apart from the ethics statement and the bit about sampling choices, which are fine).

Reviewer #3 (Remarks to the Author):

Szécényi-Nagy et al. analysed SNP capture data from 125 individuals from the Carpathian Basin, dating from the Late Neolithic to the Early Chalcolithic. They applied commonly used

population genetics tools, kinship software, and IBD analysis to investigate social organization, genetic continuity, and admixture patterns across different sites in present-day Hungary and Romania.

The authors claim that, despite differences in cultural epochs and burial practices, they found a relatively high degree of genetic continuity in the Carpathian Basin during this period. Another key finding is the lack of correlation between material culture and kinship organization systems. Two cemeteries that are close in time, space, and material culture exhibit different patterns of consanguinity, as well as variations in genetic diversity and social organization.

This study provides new insights into population transitions and human movements from the Late Neolithic to the Early Chalcolithic in the Carpathian Basin. It also contributes to our understanding of social organization during this period. The conclusions are well supported by the results, and the methodology is replicable. I appreciate how the authors support all their observations with statistical tests, something that is sometimes lacking in archaeogenetic studies. I recommend acceptance with minor changes.

Minor Comments and Suggestions

The authors conducted three contamination estimates: two based on the X chromosome (valid only for males) and one using contamMix for mitochondrial DNA. However, for the latter, only some samples have results listed in the "Assessment Note" column. I guess that the remaining samples have contamMix results close to zero. If this is the case, could the authors include a note in the Supplementary Information (SI) specifying that only samples with contamMix results above a certain threshold are annotated?

AG column in Supplementary Data 1 shows the results of ContamMix with the header "mtDNA match to consensus sequence 95% CI". The combined contamination assessment is shown in column AP-AQ. We have incorporated the detailed assessment criteria into the header of the columns AP-AQ. The threshold for mt contamination was 0.98 for the upper bounds of the confidence interval, if mitogenome coverage was above 2x.

PCA: I acknowledge that due to the high number of samples, it is difficult to display a clear figure. I find myself having trouble interpreting the PCA. Perhaps highlighting the newly reported samples or adding a zoom panel that focuses solely on the new samples, without the references, would help clarify the presentation.

We corrected the image, and highlighted the new data on the plots.

Why did you choose to use only PC1 and PC2 for detecting outliers in the main analysed groups? Would incorporating the first 10 PCs provide a more robust detection?

We tested for genetic outliers within the communities using various methods (PCA, f4, qpAdm, and IBD-based approaches). These methods consider different aspects of the genetic makeup of the communities, and their outcomes are not always consistent.

Calculating the eigenvectors in PCA, it can be observed that their values drop drastically: 7.743, 4.183, 2.526, and 1.909, and then remain low for the subsequent components. The first two components are presented in the PCA, as there is little variation in the third and fourth components. Using the Mahalanobis distance and chi-square distribution, our aim was to describe the visible outliers on the plot within a statistical framework.

Calculating outliers based on more PCs, would result in an incomprehensible set of outliers that do not correspond anymore to the observed pattern in the first two PCs.

Lines 175 and 179: Be cautious when inferring ancestry and genetic affinities based solely on PCA positions. While PCA provides useful insights into genetic relationships, it does not directly quantify ancestry or capture the complexity of population interactions. Therefore, drawing firm conclusions about genetic affinities based solely on PCA positions may oversimplify the results. I suggest rephrasing these sentences.

Our inference on HG ancestry was based on the combined observations obtained from various methods, but we accept the criticism, and rephrase these lines (174-192 in the clean document), which refer to the PCA primarily.

There is a discrepancy in the number of PCA outliers reported. In the results (lines 186–187) and Table 1 for Urziceni-Vamă, the ECA states that there are 8 PCA outliers. However, in the discussion (line 555), it mentions 11 outliers, 8 of whom are female. Should the correct figure be 8 outliers, with 5 of them being female? Please confirm.

We confirm that the first mention states the number of outliers correctly, and corrected the statement in the discussion.

To assess genetic continuity and varying degrees of admixture, the authors rely on f4-statistics, and qpAdm analyses, which are well-established methods for detecting admixture between populations. However, qpAdm results can vary significantly depending on the choice of reference populations, often producing plausible models with alternative left populations beyond those selected by the authors. Can the authors provide more detailed justification for their selection of reference (right) populations for each case and explain how their results might change if different but similar populations were used instead?

During the study design, we selected and tested various combinations of left and right populations, constating that slight changes in component proportions emerge, depending on the participants of the models.

On the first distant tests with a simple Iron Gates HG + Anatolian N (ANF) 2-way model, only 10 individuals failed out of 152 inds. With the following set of right group: Mbuti.DG, Russia_Ust_Ishim.DG, Russia_Kostenki14, England_Mesolithic, Russia_Karelia_HG, Russia_MA1_HG.SG, Ethiopia_4500BP.DG, Morocco_Iberomaurusian, Jordan_PPNB, Russia_Sidelkino_HG.SG, Russia_Arkhangelsk_Veretye_Mesolithic.SG, Georgia_Kotias.SG we aimed at covering the distant variability of those outgroups that might contributed to the source populations (left populations) in various amounts. However, whereas these tests demonstrated the distribution of the HG components, we were rather interested in the internal variation of the HG component itself, and whether we can detect any influx from the West, East or the South in the studied LN-ECA communities.

Therefore, we used EHG and WHG, knowing that Iron Gates HG represents a mixture of these two. The right set were as written in Supplementary Data 5: Mbuti.DG, Russia_Ust_Ishim.DG, Russia_Kostenki14, Russia_MA1_HG.SG, Ethiopia_4500BP.DG, Morocco_Iberomaurusian, Jordan_PPNB, Russia_Arkhangelsk_Veretye_Mesolithic.SG, Georgia_Kotias.SG, Italy_North_Villabruna_HG and Spain_EIMiron. This covered various sets of HGs from East, West and the south, minimizing also the standard error of the component estimates gained from the three left groups.

Producing still some failed tests, and having in mind the results of the Lazaridis et al. 2022 Southern Arc study, we further wanted to check for potential presence of CHG or Levant ancestry. Increasing the number of left groups however is prone to increase of the standard error, therefore, we reduced the set of right group to a smaller collection in that case: Mbuti.DG 14 Russia_Kostenki14, Russia_MA1_HG.SG, Iran_GanjDareh_N, Turkey_Epipaleolithic, Italy_North_Villabruna_HG.

Our further basic question was the testing for local continuity in the LN-ECA periods. For that, we used the GHP LN Csőszhalom and Romania N Piscoł samples in one-way models. In these tests, using Russia_EHG, Turkey_N, WHG and CHG as outliers helped us detect major additional streams whenever these tests failed.

For the group-based tests including sample sets from Bulgaria and Romania from the Penske et al. study, we decided to use the set of the right group that makes the analyses comparable to the results presented in that paper (with the same OG set used there for distal modeling: Mbuti.DG, Turkey_Epipaleolithic, Iran_GanjDareh_N, Russia_MA1_HG.SG, Russia_Kostenki14, Italy_North_Villabruna_HG). We did not compare the sites in the same analyses and did not rotate them in the tests.

In Figure 2B, could you clarify in the figure caption what the black error bar represents?

We clarified that they represent standard errors of the given qpAdm component estimates.

In the qpAdm modeling, the authors excluded one individual from each first-degree relative

pair. However, shouldn't second- and even third-degree relatives also be excluded from this analysis?

We conducted most of our analyses as individual-based tests, using all individuals who met our quality criteria (<50k SNPs on the 1.15 million autosomal targeted SNPs). In individual-based tests, relatives do not pose any issues. We identified some second-degree relatives in Basatanya, and one each in Aszód, Csőszhalom, and ECA Urziceni. However, the latter three are almost negligible compared to the size of the sample sets from each cemetery.

In Supplementary Data 5F, we present the only group-based qpAdm analysis, in which we used 24 individuals from Basatanya, 23 from Csőszhalom, and 54 from Urziceni. We did not use these groups, which were cleaned of first-degree relatives, in any group-based f_3 - f_4 statistics, as the relative affinities of the groups could be distorted by false signals arising from related individuals.

We have now repeated this test for Basatanya, additionally excluding one of each second-degree relative. The change is noticeable only in the third decimal place. We also believe that removing third-degree relatives would not be necessary in this case either.